# Token Dynamics on Spheres in Mamba Models

## Abstract

The dynamical properties of tokens in internal representations, or token dynamics, of a deep learning model has recently attracted considerable attention in deep learning theory. While transformer dynamics have been extensively studied, the analysis of token dynamics in selective state space (Mamba) models remains largely unexplored. Existing studies on Mamba models impose restrictive assumptions, such as relying solely on state space layers and limiting token embeddings to one dimension. However, practical implementations of Mamba incorporate layer normalization and operate in high dimensions, implying that token dynamics evolve on a high-dimensional unit sphere. In this work, we address this gap by formulating deep Mamba models as flow maps on high-dimensional unit spheres and providing a comprehensive theoretical analysis of their token dynamics. We characterize all possible token limit points and establish explicit exponential convergence rates toward these points. Our analysis reveals that the first token's limit point exerts an attracting effect on other tokens, leading to clustering phenomena. Furthermore, we extend our theoretical analysis of the attracting effect to a broader class of autoregressive sequence models, including state space models, causal Transformers, and classical time-series models. Leveraging this insight, we propose two applications: (i) randomly reordering tokens during training to diversify clustering on a small subset of limit points, thereby improving model performance, and (ii) explaining the attention sink effect in Mamba models through the attraction of the first token. Experimental results confirm our theoretical findings and demonstrate the practical benefits of these refinements, offering new perspectives on enhancing the effectiveness of Mamba models.

## 1 Introduction

Modern sequence modeling has been dominated by transformers, while state space models (SSMs) have emerged as an important alternative. Transformers have set the standard for representation learning, yet their quadratic complexity with respect to sequence length poses challenges for efficiency and scalability (Vaswani et al., 2017). In parallel, SSMs have reemerged as powerful alternatives that combine principles of recurrence and convolution with ideas from classical control theory (Kalman, 1960). Recent innovations in parameterization and implementation have made SSMs substantially more efficient (Gu et al., 2022b; 2021; 2022a; Gupta et al., 2022; Li et al., 2023; Kosma et al., 2023; Orvieto et al., 2023; Smith et al., 2023), enabling them to capture long-range dependencies effectively (Gu et al., 2020) and to excel in benchmark tasks such as the Long Range Arena (Tay et al., 2021). Beyond efficiency, these models have demonstrated strong empirical performance in domains as varied as audio, speech, and vision (Goel et al., 2022; Nguyen et al., 2022a; Saon et al., 2023).

Within this line of development, the Mamba architecture has emerged as a selective SSM with striking empirical results. Its distinctive component, the selective state-space (S6) layer, introduces input-dependent parameters that grant the model content awareness (Gu & Dao, 2024). This design has proven effective across a wide spectrum of applications, including language modeling (Pióro et al., 2024; Lenz et al., 2025), vision and video analysis (Liu & Li, 2024; Zhu et al., 2024; Yang et al., 2024; Li et al., 2024a), medical imaging (Ma & Wang, 2024; Wang et al., 2024b), tabular data (Ahamed & Cheng, 2024), reinforcement learning (Ota, 2024), 3D perception (Liang et al., 2025), graph learning (Wang et al., 2024a), and even $N$-dimensional sequence modeling (Li et al., 2024b). The breadth of these successes has been linked to Mamba's ability to generate highly adaptive and expressive representations across modalities (Rezaei Jafari et al., 2025; Patro & Agneeswaran, 2024; Xu et al., 2024; Zhu et al., 2024).

The study of token dynamics–the evolution of token representations within a model–has proven to be a powerful framework for understanding transformers, offering insights into equilibrium structures, stability properties, and clustering phenomena. In contrast, the token dynamics of selective SSMs have received little attention, despite the strong empirical performance of architectures such as Mamba. Existing analyses rely on restrictive assumptions, such as one-dimensional embeddings or the exclusion of normalization layers, which substantially oversimplify the behavior of practical implementations Vo et al. (2025). Real-world Mamba models operate in high-dimensional embedding spaces and incorporate layer normalization, leading to internal dynamics that are considerably richer than current theories capture.

## 1.1 OUR CONTRIBUTIONS

In this paper, we investigate the dynamical properties of tokens in deep pretrained Mamba models under more realistic and practically relevant assumptions on model parameters. Prior theoretical studies often rely on restrictive conditions such as the absence of layer normalization and one-dimensional token embeddings (see (Vo et al., 2025)). We consider the continuous-time limit of pretrained Mamba models without these restrictive assumptions and systematically analyze the asymptotic behavior of token trajectories over time. Our main contributions are summarized as follows:

- We derive a dynamical system governing the continuous-time limit of pretrained deep Mamba models. In contrast to previous studies, our analysis incorporates layer normalization and investigates token dynamics on high-dimensional spheres.

- We characterize the set of limit points of token trajectories in Mamba models and establish explicit exponential convergence rates. Our theoretical analysis further demonstrates that the first token exerts an attracting influence on the remaining tokens, a phenomenon that we empirically validate.

- We extend our theoretical analysis of the attracting effect of the first token on the remaining tokens to a broader class of autoregressive sequence models, including state space models, causal Transformers, and classical time-series models. We then strengthen a result established in (Karagodin et al., 2024) about the asymptotic convergence of tokens in causal attention dynamic.

- We demonstrate the practical implications of our theoretical results through two applications: (i) offering a principled explanation of the attention sink effect in Mamba models via the attracting role of the first token, and (ii) randomizing token order during training to diversify clustering.

To validate our theoretical findings and their applications, we conduct extensive experiments by visualizing attention concentration in Falcon-Mamba-7B across different depths and by testing the impact of random token reordering in MambaVision-T on ImageNet-1K classification benchmarks. These experimental results confirm the predictive accuracy of our analysis and underscore its practical benefits in real-world scenarios.

## 2 MAMBA DYNAMIC AS A FLOW MAP ON SPHERE AND OBSERVATIONS

### 2.1 BACKGROUND – S6 LAYER

The building block of a selective state-space (Mamba) model is a Mamba block (see (Dao & Gu, 2024)). At the heart of the Mamba block is the S6 layer, which plays a key role on the success of deep Mamba models.

S6 **layer.** Formally, an S6 layer is defined as a function that maps a sequence of tokens $\mathbf{x} = (x_1, \ldots, x_L) \in \mathbb{R}^{D \times L}$ to another sequence of tokens $\mathbf{y} = (y_1, \ldots, y_L) \in \mathbb{R}^{D \times L}$ (with the same number of channels $D$). In this context, each vector $x_l \in \mathbb{R}^D$ (or $y_l \in \mathbb{R}^D$) represents a token/word, while the whole sequence $\mathbf{x} = (x_1, \ldots, x_L)$ (or $\mathbf{y} = (y_1, \ldots, y_L)$) is called a prompt/sentence.

For each $d = 1, \ldots, D$, the $d$-th channel output $\hat{y}_d = (y_{d1}, \ldots, y_{dL}) \in \mathbb{R}^L$ is determined recurrently from the corresponding $d$-th channel input $\hat{x}_d = (x_{d1}, \ldots, x_{dL}) \in \mathbb{R}^L$ via a sequence of hidden states $h_{d1}, \ldots, h_{dL} \in \mathbb{R}^N$ as follows:

$$\begin{cases} h_{dl} = \overline{A}_{dl} \cdot h_{d,l-1} + \overline{B}_{dl} \cdot x_{dl}, & h_{d0} = 0, \\ y_{dl} = C_l \cdot h_{dl}, \end{cases} \tag{1}$$

for $l = 1, \ldots, L$. Here, the matrices $\overline{A}_{dl}$, $\overline{B}_{dl}$, and $C_l$ are input-dependent and time-varying, determined by

$$\overline{A}_{dl} = e^{-a_d \Delta_d(x_l)}, \quad \overline{B}_{dl} = \Delta_d(x_l) S_B \cdot x_l, \quad C_l = (S_C \cdot x_l)^\top, \tag{2}$$

where $\Delta_d : \mathbb{R}^D \to \mathbb{R}$ is the step size function at the $d$-th channel and is defined by

$$\Delta_d(u) = \text{softplus}(S_{\Delta,d} \cdot u) = \ln\left(1 + e^{S_{\Delta,d} \cdot u}\right), \qquad u \in \mathbb{R}^D. \tag{3}$$

The hidden positive coefficients $a_d \in \mathbb{R}_{>0}$, and the step size vectors $S_{\Delta,d} \in \mathbb{R}^D$ for $d = 1, \ldots, D$, as well as the input and output matrices $S_B, S_C \in \mathbb{R}^{N \times D}$, are learnable from input data.

By eliminating the hidden states $h_{dl}$ in system (1) and recall that $y_l = (y_{1l}, \ldots, y_{DL})$, we obtain the following form (see also (Ali et al., 2024)):

$$y_l = \sum_{j=1}^{l} A_{lj}(\mathbf{x}) x_j, \tag{4}$$

where $A_{lj} = \text{diag}(A_{1,lj}, \ldots, A_{D,lj})$ and $S_{BC} = S_C^\top S_B$ as well as

$$A_{d,lj}(\mathbf{x}) = \begin{cases} x_l^\top S_{BC} x_l \cdot \Delta_d(x_l), & \text{if } l = j, \\ x_l^\top S_{BC} x_j \cdot \Delta_d(x_j) \cdot \exp\left(-a_d \sum_{k=j+1}^{l} \Delta_d(x_k)\right), & \text{if } l > j. \end{cases} \tag{5}$$

In our context, the matrix $S_{BC} = S_C^\top S_B \in \mathbb{R}^{D \times D}$, which we will refer to as the *input-output matrix*, plays an essential role in characterizing the dynamical properties of tokens.

## 2.2 MAMBA DYNAMIC AS FLOW MAP ON SPHERE

All practical implementations of Mamba models incorporate layer normalization (Ba et al., 2016), most commonly realized as root mean square (RMS) normalization (Zhang & Sennrich, 2019). In this approach, the sequence of tokens at each layer is normalized by dividing each token by its Euclidean norm. This operation enhances stability by preventing divergence, thereby reducing rounding errors and avoiding numerical overflow. Consequently, the resulting dynamics can be viewed as evolving on the unit sphere $\mathbb{S}^{D-1}$ throughout.

Follow the framework of (Karagodin et al., 2024), a Mamba model including S6 layer and layer normalization with skip connection can then be viewed as a flow map on $(\mathbb{S}^{D-1})^L$ of the dynamical system

$$\frac{d}{dt} x_l(t) = \mathbf{P}_{x_l(t)} \left( \sum_{j=1}^{l} A_{lj}(\mathbf{x}(t), t) x_j(t) \right), \quad l = 1, \ldots, L, \tag{6}$$

$$\mathbf{x}(0) = (x_1(0), \ldots, x_L(0)) \in (\mathbb{S}^{D-1})^L, \tag{7}$$

where

- for each $x \in \mathbb{S}^{D-1}$, the map $\mathbf{P}_x : \mathbb{R}^D \to T_x \mathbb{S}^{D-1}$ is the projection onto the tangent space at $x$ of the sphere $\mathbb{S}^{D-1}$ and it is defined by $\mathbf{P}_x(y) = y - \langle y, x \rangle x$ for each $y \in \mathbb{R}^D$;

- the coefficient $A_{lj}$ is an $D \times D$ diagonal matrix $A_{lj} = \text{diag}(A_{1,lj}, \ldots, A_{D,lj})$ with

$$A_{d,lj}(\mathbf{x}(t), t) = \begin{cases} x_l(t)^\top S_{BC}(t) x_l(t) \cdot \Delta_d(x_l(t), t), & \text{if } l = j, \\ x_l(t)^\top S_{BC}(t) x_j \cdot \Delta_d(x_j(t), t) \cdot \exp\left(-a_d(t) \sum_{k=j+1}^{l} \Delta_d(x_k(t), t)\right), & \text{if } l > j; \end{cases} \tag{8}$$

- the step size function at the $d$-th channel $\Delta_d : \mathbb{R}^D \times [0, +\infty) \to \mathbb{R}$ is defined by $\Delta_d(u, t) = \text{softplus}(S_{\Delta,d}(t)u)$, for each $(u, t) \in \mathbb{R}^D \times [0, +\infty)$.

**Remark 2.1** (Parameters are time-dependent in simulations and experiments). Dynamical system (6) is parametrized by the input-output matrix $S_{BC}(t) = S_C(t)^\top S_B(t) \in \mathbb{R}^{D \times D}$, the step-size matrix $S_\Delta(t) = [S_{\Delta,1}(t), \ldots, S_{\Delta,D}(t)] \in \mathbb{R}^{D \times D}$, and the state coefficient vector $(a_1(t), \ldots, a_D(t)) \in \mathbb{R}^D_{>0}$. All of these parameters are time-dependent, as they may vary across different layers. Accordingly, we will perform simulations and experiments in the subsequent sections using time-dependent parameters in Sections 2.3 and 4. However, for the sake of rigorous analysis, we will derive theoretical results under the assumption of time-independent parameters in Section 3.

**Remark 2.2** (Theoretical justification and practical relevance of the continuous-time limit). The dynamical system (6) captures the continuous-time limit of deep Mamba models. Since modern deep networks often comprise a very large number of layers, they can be effectively approximated by continuous-time dynamical systems. A notable example is the interpretation of ResNets as discretizations of ordinary differential equations (ODEs), an approximation that becomes increasingly accurate as network depth grows. The continuous-time perspective is a well-established and widely adopted approach in both the machine learning and mathematics communities.

Our main problem is studying the dynamical properties of the solutions of system (6). In particular, we want to understand how the set $\lim_{t \to +\infty} \{x_1(t), \ldots, x_L(t)\}$ looks like and when it reduces to a set of a single point.

## 2.3 Observations from Simulations with Random Parameters

Before providing a rigorous theoretical analysis of the token dynamics of Mamba via dynamical system (6), we run several simulation with random choices of model parameters to have some observations of the results. In particular, we numerically integrate the system to observe: (i) trajectories of first tokens across many random initializations with time-independent parameters $S_{BC}$ and $S_\Delta$; (ii) trajectories of all tokens in a single sequence with time-independent parameters $S_{BC}$ and $S_\Delta$; and (iii) trajectories of all tokens in a single sequence with time-dependent parameters $S_{BC}$ and $S_\Delta$. Figure 1 demonstrates the expected results of the trajectories of the tokens under these settings.

**Observation 2.3.** From Figure 1, two consistent patterns can be observed:

   ($i$) The first token has privileged dynamics: its trajectory determines a discrete set of possible limit points, primarily the coordinate unit vectors $\pm e_d$, which serve as attractors. Here, $e_d$ is the coordinate unit vector with 1 at the $d$-th coordinate and 0 elsewhere.

   ($ii$) Subsequent tokens are strongly influenced by the first token, often converging toward the same attractor, thereby creating clusters of tokens in the final layers.

The above observation directly supports the claim that the first token acts as a global attractor and explains phenomena like the attention sink effect in Mamba/causal models (see Section 4.1 for further details). The next section provides a theoretical explanation for the above observation.

## 3 Theoretical Justifications

In this subsection, we provide a theoretical justification for Observation 2.3. In particular, we characterize the possible limit points of the first token in a pretrained deep Mamba model in Subsection 3.1, prove that all other tokens will exponentially converge to certain attractors corresponding to the limit points of the first token in Subsection 3.2, and extend our convergence results beyond Mamba, to general autoregressive sequence models in Subsection 3.3.

The study of the convergence of all tokens based on the dynamical system (6) without any additional assumption on the model parameters is challenging. To make the proofs feasible while still maintaining the generality of our theoretical findings to verify Observation 2.3, in this whole section, we will assume that all parameters $S_{BC}, S_{\Delta,d}$ and $a_d$ are time-independent.

### 3.1 Possible Limit Points of the First Token

Let us consider the differential equation governing the trajectory of the first token, which is the first equation in the dynamical system (6), given by

$$\frac{d}{dt} x_1(t) = \mathbf{P}_{x_1(t)} \big( A_{11}(\mathbf{x}(t)) x_1(t) \big), \tag{9}$$

$$x_1(0) \in \mathbb{S}^{D-1},$$

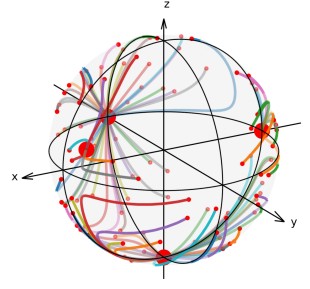 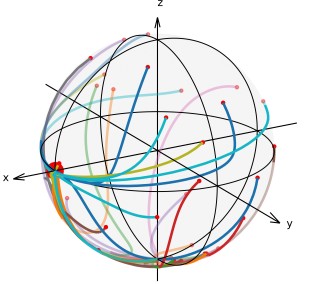 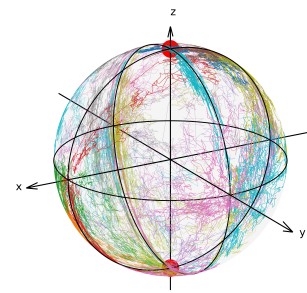

(a) First-token trajectories (100 random initializations; time-independent $S_{BC}$, $S_\Delta$). Most trajectories converge to the coordinate unit vector $\pm e_d$ for some $d$.

(b) A single sequence of 32 tokens (time-independent $S_{BC}$, $S_\Delta$). Tokens converge toward a point at one coordinate unit vector $\pm e_d$ for some $d$.

(c) A single sequence of 16 tokens (time-dependent $S_{BC}$, $S_\Delta$). Tokens converge to points at some coordinate unit vectors $\pm e_d$ for some $d = 1, \ldots, D$.

Figure 1: Simulated trajectories of dynamical system (6) with random parameters. In these settings, the first token converges to discrete attractors (mostly coordinate unit vectors $\pm e_d$) and draws subsequent tokens toward the same points, producing clusters and illustrating the attention sink effect.

where

$$A_{11} = \text{diag}(A_{1,11}, A_{2,11}, \cdots, A_{D,11}), \quad \text{and} \quad A_{d,11}(\mathbf{x}(t)) = x_1(t)^\top S_{BC} x_1(t) \cdot \Delta_d(x_1(t)).$$

A limit point must be an equilibrium point. We will determine the set of equilibrium points of differential equation (9) in the following proposition.

**Proposition 3.1** (All possible equilibrium points of the first token). *A vector $y \in \mathbb{S}^{D-1}$ is an equilibrium of equation (9) if and only if it belongs to one of the following types:*

> ***Type 1.*** $y \in \bigcup_{A \subset \{1,2,\ldots,D\}} \mathcal{H}_{\Delta,A}$, *where for each subset $A \subset \{1, \ldots, D\}$, we denote by*
> $\mathcal{H}_{\Delta,A} := \langle \{ S_{\Delta,j} - S_{\Delta,\min A}, \, j \in A \} \cup \{e_k, \, k \in A\} \rangle^\perp$.

> ***Type 2.*** $y \in \mathcal{H}_{S_{BC}} := \{u \in \mathbb{R}^D \,|\, u^\top S_{BC}\, u = 0\}$, *which is the null space of the quadratic surface defined by $S_{BC}$.*

*If $S_{BC}$ is definite, then $\mathcal{H}_{S_{BC}} = \emptyset$ and Type 2 equilibria do not exist. If, furthermore, the columns of $S_\Delta$ are linearly independent, then the Type 1 equilibria collapse to exactly two antipodal points $y = \pm\xi$, where $\xi$ is the unique unit vector in the union $\bigcup_{A \subset \{1,2,\ldots,D\}} \mathcal{H}_{\Delta,A}$.*

Even for the first token, the study of stability near equilibrium points in order to identify the limit points of equation (9) is challenging and technically involved. We leave the proof of the above proposition, as well as the study of the stability of these equilibrium points, to Appendix B.1.1.

In Observation 2.3 $(i)$, we observe that the first token likely converge to a limit point of the form $\pm e_d$ for some $d = 1, \ldots, D$. The following theorem provide a justification for this observation.

**Theorem 3.2** (A justification for Observation 2.3 $(i)$ – Convergence to $\pm e_d$ of the first token). *Let $y(t) = (y_1(t), \ldots, y_D(t)) \in C^\infty([0, +\infty), \mathbb{S}^{D-1})$ be a solution of differential equation (9). Assume that the symmetric part of $S_{BC}$ is positive definite, and $S_{\Delta,m} = \text{sign}(y_m(0)) \cdot e_m$ for each $m$ such that $y_m(0) \neq 0$. Then*

$$\lim_{t \to \infty} y(t) = \text{sign}(y_d(0)) \cdot e_d,$$

*where $d$ is the index such that $|y_d(0)| = \max_{1 \leq i \leq D} |y_i(0)|$.*

In the two-dimensional case, we can actually determine all possible limit points of the first token in the following theorem.

**Theorem 3.3** (All possible limit points of the first token in two-dimension). *Assume that $D = 2$ and let $y(t) = (y_1(t), y_2(t)) \in C^\infty([0, +\infty), \mathbb{S}^1)$ be a solution of differential equation (9). Assume further that*

> *(i) $S_{BC} = \text{diag}(\lambda_1, \lambda_2)$ with $\lambda_1 \neq \lambda_2$ or $\lambda_1 = \lambda_2 \neq 0$; and*

*(ii)* $S_{\Delta,2} \neq S_{\Delta,1}$.

*In this case, the circle $\mathbb{S}^1$ intersects the line $\langle S_{\Delta,2} - S_{\Delta,1}\rangle^\perp$ at two antipodal points, say $\pm\xi$. Then $\lim_{t\to+\infty} y(t)$ exists and it belongs to the set*

$$\Omega = \{(0,\pm 1), (\pm 1, 0)\} \cup \{\pm\xi\} \cup \left\{ e^{i\varphi_\star} : \cos(2\varphi_\star) = \frac{\lambda_1 + \lambda_2}{\lambda_1 - \lambda_2} \right\}.$$

**Remark 3.4** (Possible limit points of the first token). Theorem 3.2 indicates that the first token converges to one of the coordinate unit vectors $\pm e_m$, consistent with Observation 2.3. In contrast, Theorem 3.3 reveals additional possible limit points, which are difficult to detect through simulations with random choices of model parameters.

## 3.2 ALL OTHER TOKENS ARE ATTRACTED BY THE FIRST TOKEN

Different from the first token, the problem of characterizing the possible limits points of the other token is far more challenge. In this subsection, to make the proofs feasible while still maintaining the generality of our findings, beside of the time-independent parameters assumption, we will assume further that:

$$S_{\Delta,1}(t) = \ldots = S_{\Delta,D}(t) =: \alpha \in \mathbb{R}^D \quad \text{and} \quad a_1(t) = \ldots = a_D(t) =: a \in \mathbb{R}, \qquad (10)$$

for any time $t \in [0, +\infty)$. Under this assumption, we can rewrite dynamical system (6) as:

$$\frac{d}{dt} x_l(t) = \sum_{j=1}^{l-1} A_{lj}(\mathbf{x}(t)) \cdot \mathbf{P}_{x_l(t)}(x_j(t)), \quad l = 1, \ldots, L, \qquad (11)$$

$$\mathbf{x}(0) = (x_1(0), \ldots, x_L(0)) \in (\mathbb{S}^{D-1})^L,$$

where the coefficients $A_{lj}(\mathbf{x}(t))$ are defined as

$$A_{lj}(\mathbf{x}(t)) = x_l(t)^\top S_{BC} x_j(t) \cdot \Delta(x_l(t)) e^{-a \sum_{k=j+1}^{l} \Delta(x_k(t))}, \qquad (12)$$

with $\Delta(x_l(t)) = \ln(1 + e^{\alpha \cdot x_l(t)})$.

In this case, for the first token $x_1(t)$, we see that

$$\frac{d}{dt} x_1(t) = A_{11}(\mathbf{x}(t), t) \cdot \mathbf{P}_{x_1(t)}(x_1(t)) = 0.$$

This implies that $x_1(t) = x_1(0)$ is a constant function on $[0, +\infty)$. We will present all possible limit points of $x_i(t)$ for $i = 2, \ldots, L$ in Theorem 3.5 and Proposition 3.6 below.

**Theorem 3.5** (A justification for Observation 2.3 *(ii)* – First token exponentially attracts other tokens). *Let $\mathbf{x}(t) = (x_1(t), \ldots, x_L(t)) \in C^\infty([0, +\infty), (\mathbb{S}^{D-1})^L)$ be a solution of dynamical system (11). Assume that $S_{BC}$ is symmetric. Assume further that $\langle x_1(0), S_{BC} x_1(0)\rangle > 0$, and for $i \geq 2$,*

$$\langle x_i(0), S_{BC} x_1(0)\rangle \notin \{0, \pm 1\}, \quad \langle x_i(0), x_1(0)\rangle \notin \{\pm 1\}.$$

*Then, there exists a zero-measurable subset $\Omega \subset \mathbb{S}^{D-1}$ such that: if the initial points $x_l(0) \notin \Omega$ for all $l$, then all tokens converge exponentially to $\pm x_1(0)$, i.e*

$$\sum_{i=1}^{L} |x_i(t) - s_i x_1(0)| \lesssim e^{-C_0 t},$$

*for some constant $C_0 > 0$ and $s_l \in \{+1, -1\}$ for each $l$.*

It is natural to ask under what conditions the convergence of all tokens in Theorem 3.5 above is guaranteed for arbitrary initial states. The following proposition establishes that a sufficient condition is the initial positivity of all token interactions.

**Proposition 3.6** (Exponential alignment under positive angle conditions). *Suppose the assumptions of Theorem 3.5 hold. In addition, assume that the initial configuration satisfies the positive angle condition:*

$$\langle x_i(0), S_{BC} x_j(0)\rangle > 0, \quad \forall i, j \in [1, L].$$

*Then all tokens $x_l(t)$ converge exponentially to the aligned state $x_1(0)$, that is,*

$$\sum_{i=1}^{L} \|x_i(t) - x_1(0)\| \lesssim e^{-C_0 t},$$

*for some constant $C_0 > 0$. Moreover, the positivity of interactions is preserved for all time:*

$$\langle x_i(t), S_{BC} x_j(t) \rangle > 0, \quad \forall t > 0.$$

**Remark 3.7** (First token exponentially attracts other tokens). Theorem 3.5 together with Proposition 3.6 implies that all tokens converge exponentially to the limit points of the first token, thereby confirming Observation 2.3 (ii). This, in turn, indicates that tokens in a deep Mamba model tend to form a small number of clusters in the final layers.

### 3.3 BEYOND MAMBA: AUTOREGRESSIVE SEQUENCE MODELS

Our analysis of the attracting effect of the first token on the remaining tokens extends to a broader class of autoregressive sequence models, including state space models, causal Transformers, and classical time-series models. In particular, consider the dynamical system

$$\frac{d}{dt} x_l(t) = \sum_{j=1}^{l-1} a_{lj}(\mathbf{x}(t)) \cdot \mathbf{P}_{x_l(t)}(x_j(t)), \quad l = 1, \ldots, L, \tag{13}$$

$$\mathbf{x}(0) = (x_1(0), \ldots, x_L(0)) \in (\mathbb{S}^{D-1})^L,$$

where the coefficients $a_{lj} = a_{lj}(x_1(t), \ldots, x_l(t))$ are positive and continuously differentiable functions.

**Remark 3.8** (Equation (13) as the continuous-time limit of autoregressive sequence models). The dynamical system (13) can be interpreted as the continuous-time limit of various deep autoregressive sequence models. In particular, it reduces to the continuous-time limit of a deep Mamba models, as described in equation (11), when the coefficients are chosen as $a_{lj} = A_{lj}$ defined in equation (12). On the other hand, the same system specializes to the continuous-time limit of causal attention when the coefficients are chosen as

$$a_{lj} = \frac{e^{\langle Q x_l(t), K x_j(t) \rangle}}{\sum_{i=1}^{l} e^{\langle Q x_l(t), K x_i(t) \rangle}},$$

where $Q$, $K$, and $V$ denote the query, key, and value matrices, respectively (see Karagodin et al. (2024)).

For this general setting we prove that:

**Theorem 3.9** (Convergence in autoregressive sequence models). *Let $\mathbf{x}(t) = (x_1(t), \ldots, x_L(t)) \in C^\infty([0, +\infty), (\mathbb{S}^{D-1})^L)$ be a solution of dynamical system (13). Assume that $\langle x_l(0), x_1(0) \rangle \notin \{-1, 1\}$ for all $l \geq 2$. Then, there exists a zero-measurable subset $\Omega \subset \mathbb{S}^{D-1}$ such that: if the initial points $x_l(0) \notin \Omega$ for all $l$, then all tokens converge exponentially to $\pm x_1(0)$, i.e*

$$\sum_{i=1}^{L} |x_i(t) - x_1(0)| \lesssim e^{-C_0 t},$$

*for some constant $C_0 > 0$.*

**Remark 3.10** (Strengthening of a result in (Karagodin et al., 2024)). By Theorem 3.9, we not only recover the asymptotic convergence established in (Karagodin et al., 2024, Theorem 4.1), but also prove a stronger quantitative result: the convergence takes place at an exponential rate. This refinement sharpens the classical theory and yields a more complete characterization of the single clustering case in a general setting of autoregressive sequence models.

## 4 APPLICATIONS – FIRST TOKEN MATTERS

Observation 2.3, together with the theoretical justification provided in the previous section, suggests that the first token should be regarded with particular importance, as it serves as an attractor for subsequent tokens. Building on this insight, we provide in Subsection 4.1 a principled explanation of the attention sink effect in Mamba models through the attracting role of the first token. Moreover, in Subsection 4.2, we propose a practical strategy for Mamba models: randomly reordering tokens during training to diversify clustering around a small set of limit points, thereby enhancing model performance. We provide the details of experiment setup and evaluation benchmark of both experiments in Appendix C.

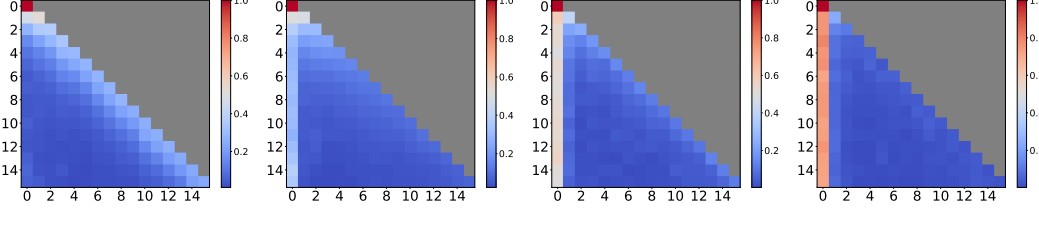

(a) Layer 1, Channel 8    (b) Layer 15, Channel 8    (c) Layer 46, Channel 8    (d) Layer 60, Channel 8

Figure 2: Attention concentration on the first token across different depths of the Falcon-Mamba-7B model (Zuo et al., 2024). Each subfigure shows the hidden attention matrix for a specific layer: (a) Layer 1, (b) Layer 15, (c) Layer 46, and (d) Layer 60. In early layers, tokens distribute attention more uniformly, but as depth increases, nearly all tokens concentrate disproportionately on the first token. This illustrates the emergence of an attention sink in state-space models, analogous to the phenomenon previously observed in transformer models.

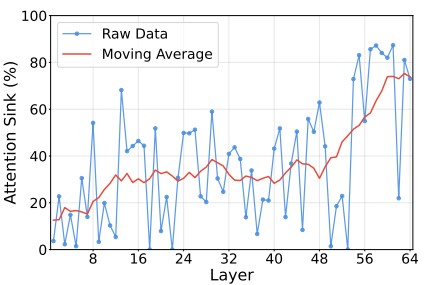 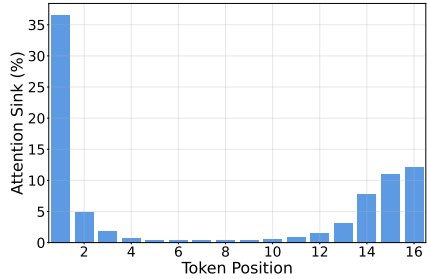

(a) Attention sink (%) across layers for the first token.

(b) Attention sink (%) across all token positions.

Figure 3: Attention sink score (Gu et al., 2025) analysis in the Falcon-Mamba-7B model. Subfigure (a) shows the sink score of the first token and its moving average across layers; the score steadily increases with depth and dominates in the final layers. Subfigure (b) shows the average sink scores across token positions; the first token consistently receives the largest share.

### 4.1 PREDICT ATTENTION SINK

We analyze the hidden attention matrix in equation (4) for the Falcon-Mamba-7B model (Zuo et al., 2024) to quantify how tokens attend to one another. The results in Figure 2 reveal that the concentration of attention on the first token consistently increases with depth, and in the final layers it dominates nearly all other tokens. This empirical trend validates our theoretical justification for token dynamics and, importantly, represents the first observation of the attention sink phenomenon in state-space models. While attention sink was originally identified in transformer-based architectures (Xiao et al., 2024), we have showed that an analogous effect emerges in Mamba models as predicted from Observation 2.3.

To quantify the attention sink in state-space models, we employ the metrics $\text{Sink}_\ell$ and $\text{Sink}$ (formally defined in Appendix C.1), adapted from (Gu et al., 2025) with modifications to account for the structural differences in our hidden attention representation. Specifically, while the original metric is computed from attention logits with shape (batch, head, seqlen, seqlen), our Mamba hidden attention has shape (batch, channel, seqlen, seqlen). Each channel plays an analogous role to an attention head, but given the large number of channels (4096), we uniformly sample 200 channels per layer for evaluation. We fix the threshold to $\varepsilon = 0.3$ to decide whether the first token exhibits sink in a given channel. The results, shown in Figure 3, reveal that the sink score of the first token increases consistently with depth and that, when averaged across token positions, the first token dominates all others, thereby extending the attention sink phenomenon to state-space models.

### 4.2 RANDOMLY REORDERING TOKENS TO DIVERSIFY CLUSTERING EFFECT

In vision models, each input sequence can be expressed as $x = (x_1, \ldots, x_L) \in \mathbb{R}^{D \times L}$, where each token $x_i \in \mathbb{R}^D$ corresponds to a patch extracted from the image. Unlike natural language, the ordering of $\{x_i\}_{i=1}^L$ does not carry semantic meaning. Nevertheless, our theoretical analysis in Section 3 shows that the evolution of the Mamba dynamical system (6) is dominated by the trajectory of the first token $x_1(t)$. Its limit point acts as an attractor for $\{x_i(t)\}_{i=2}^L$, which implies that, even

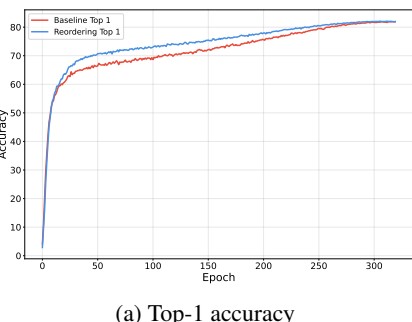 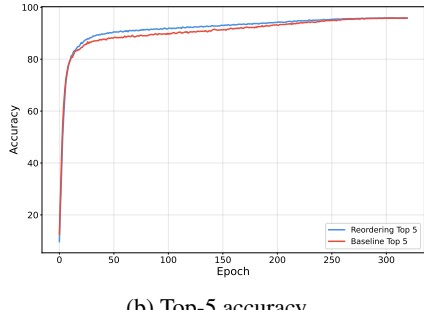

(a) Top-1 accuracy                    (b) Top-5 accuracy

Figure 4: Comparison of training accuracy between baseline MambaVision-T and our proposed random token reordering strategy on ImageNet-1K. (a) shows top-1 accuracy and (b) shows top-5 accuracy across 300 epochs. In both cases, the reordering strategy consistently outperforms the baseline, confirming that diversifying the first token embedding improves generalization and validates our theoretical framework.

in the absence of an inherent sequential order, the choice of the first token exerts a disproportionate influence on the overall token dynamics.

Motivated by this observation, we propose a random reordering strategy, in which a random permutation is applied to the input sequence before each forward pass through the Mamba block. As a result, the identity of the first token varies across passes, ensuring that the attractor role is not consistently assigned to a fixed embedding. This diversification prevents collapse toward a narrow set of limit points and instead encourages the dynamics to explore a richer subset of the representation space.

We evaluate this approach on the ImageNet-1K dataset (Deng et al., 2009) using MambaVision-T (Hatamizadeh & Kautz, 2024) as the baseline. Top-1 and Top-5 classification accuracy are reported in Table 1 and Figure 4. The results demonstrate that the reordering scheme improves performance and again provide empirical validation of our theoretical claim.

Table 1: Top-1 accuracy (%) and Top-5 accuracy (%) of MambaVision-T with and without random token reordering on ImageNet-1K classification. Random reordering diversifies the embedding of the first token, validating our theoretical framework and leading to consistent improvements. For all metrics, a higher value is better.

| Model | Top-1 (%) ↑ | Top-5 (%) ↑ |
|---|---|---|
| MambaVision-T (baseline) | 81.90 | 95.86 |
| MambaVision-T + Random Reordering | **82.09** | **95.92** |

## 5 CONCLUSION AND LIMITATION

In this work, we developed a theoretical framework for token dynamics in deep Mamba models under realistic assumptions. Modeling Mamba as flow maps on high-dimensional spheres, we characterized limit points of token trajectories, proved exponential convergence, and revealed the first token's role as a global attractor. We further extended our analysis to broader autoregressive models and demonstrated practical implications, including an explanation of the attention sink effect and a refinement via random token reordering. These results advance the understanding of token dynamics and suggest new directions for improving Mamba models.

Our analysis focuses on the continuous-time limit and asymptotic behavior, and future work is needed to fully bridge these insights with the discrete, large-scale implementations of Mamba models in practice.

**Ethics Statement.** Given the nature of the work, we do not foresee any negative societal and ethical impacts of our work.

**Reproducibility Statement.** Source codes for our experiments are provided in the supplementary materials of the paper. The details of our experimental settings and computational infrastructure are given in Section 4 and the Appendix. All datasets that we used in the paper are published, and they are easy to access in the Internet.

**LLM Usage Declaration.** We use large language models (LLMs) for grammar checking and correction.

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

# Appendix of "Token Dynamics on Spheres in Mamba Models"

**Table of Contents**

## A  RELATED WORKS

**ResNets and Neural ODEs**  Our work connects to the literature that interprets deep neural networks (DNNs) through the lens of dynamical systems. Standard DNNs can be seen as discrete-time dynamical processes, where data propagate sequentially through layers (LeCun et al., 2015). Residual networks (ResNets) (Chen et al., 2018; Haber & Ruthotto, 2017; Kpotufe & Boularias, 2012) exemplify this view, and their continuous counterparts, neural ordinary differential equations (neural ODEs) (Lin & Jegelka, 2018; Zhang et al., 2020; Li et al., 2022; Nguyen et al., 2022b; Ruiz-Balet & Zuazua, 2023; Cheng et al., 2023), formalize the passage to continuous time. In this formulation, the residual block update $z = x + y(x, \theta)$ becomes a flow governed by a differential equation, mapping an initial input $x(0)$ to an output $x(T)$. A broad line of work has studied the approximation, interpolation, and controllability properties of these architectures (Lin & Jegelka, 2018; Zhang et al., 2020; Li et al., 2022; Tabuada & Gharesifard, 2022; Ruiz-Balet & Zuazua, 2023; Cheng et al., 2023).

**Oversmoothing and Rank Collapse**  Transformers face two critical degeneracies: oversmoothing and rank collapse. Oversmoothing causes token representations to become indistinguishable, leading to a loss of expressive power. Wu et al. Wu et al. (2024), Dovonon et al. Dovonon et al. (2024), and Chen et al. Chen et al. (2022) proved that self-attention alone cannot prevent this collapse. Remedies include *ContraNorm*, a normalization method inspired by contrastive learning (Guo et al., 2023), and the use of residual connections with normalization, which have been shown to mitigate oversmoothing (Scholkemper et al., 2024).

Rank collapse, on the other hand, occurs when the hidden representations degenerate into a low-dimensional subspace. Prior studies (Dong et al., 2021; Feng et al., 2022; Noci et al., 2022; Joudaki et al., 2023; Zhai et al., 2023; Noci et al., 2024; Bao et al., 2024) identified this as a major bottleneck and proposed stabilization strategies to maintain diversity and improve performance.

**Theoretical Analyses of Mamba**  Recent studies have begun to explore Mamba from a theoretical standpoint. Ali et al. Ali et al. (2024) interpreted the S6 layer as a linear-time, softmax-free variant of attention, proving that Mamba is more expressive than transformers. In contrast, Muca Cirone et al. Muca Cirone et al. (2025) applied Rough Path Theory to show that diagonal selective state space models (SSMs) are less expressive than non-diagonal ones. Merrill et al. Merrill et al. (2024) used circuit complexity theory to argue that both SSMs and transformers belong to the class $TC^0$, implying computational equivalence. Jelassi et al. Jelassi et al. (2024) provided both theoretical and empirical evidence that transformers surpass state space models in copying tasks. Together, these results form a theoretical basis for understanding the expressive and generalization limits of Mamba.

**Token Dynamics and Clustering Effects**  Another perspective interprets sequence models, such as Transformers and Mamba, as interacting particle systems. Lu et al. Lu et al. (2019) formulated self-attention as a numerical solver for a convection–diffusion equation, while Dutta et al. Dutta et al. (2021) proposed *TransEvolve*, a particle-inspired scheme that reduces complexity. These approaches connect Transformer dynamics to broader nonlinear systems, including flocking (Ha et al., 2019), synchronization (Kuramoto, 1975; Acebrón et al., 2005), and consensus models (Krause et al., 2000; Motsch & Tadmor, 2014).

From this dynamical viewpoint, the long-term behavior of tokens has been studied via clustering. Geshkovski et al. Geshkovski et al. (2024b) proved convergence to limit objects influenced by initial conditions, while follow-up work (Geshkovski et al., 2023; 2024a) showed metastable multi-cluster states that persist for long times before collapsing into a single cluster. Alcalde et al. Alcalde et al. (2024) further confirmed asymptotic convergence to clustered equilibria in hardmax Transformers. In contrast, the token dynamics of Mamba models remain largely unexplored. Existing analyses impose restrictive assumptions - such as one-dimensional embeddings or the absence of normalization layers - that oversimplify practical implementations Vo et al. (2025).

In this paper, we take one step further by studying the token dynamics for Mamba without these restrictive assumptions.

## B  THEORETICAL RESULTS

Recall that a pre-trained Mamba model can be viewed as a flow map on $(\mathbb{S}^{D-1})^L$ defined by the following dynamical system:

$$\frac{d}{dt}x_l(t) = \mathbf{P}_{x_l(t)}\left(\sum_{j=1}^{l} A_{lj}(\mathbf{x}(t))x_j(t)\right), \quad t > 0, \, l = 1, \ldots, L, \tag{14}$$

$$\mathbf{x}(0) = (x_1(0), \ldots, x_L(0)) \in (\mathbb{S}^{d-1})^L, \tag{15}$$

where

- for each $x \in \mathbb{S}^{D-1}$, the map $\mathbf{P}_x \colon \mathbb{R}^D \to T_x\mathbb{S}^{D-1}$ is the projection onto the tangent space at $x$ of the sphere $\mathbb{S}^{D-1}$ and it is defined by

$$\mathbf{P}_x(y) = y - \langle y, x\rangle\, x, \quad y \in \mathbb{R}^D;$$

- the coefficient matrices $A_{lj}$ is an $D \times D$ diagonal matrix $A_{lj} = \text{diag}(A_{1,lj}, \ldots, A_{D,lj})$ with

$$A_{d,lj}(\mathbf{x}(t)) = \begin{cases} x_l(t)^\top S_{BC} x_l(t) \cdot \Delta_d(x_l(t)), & \text{if } l = j, \\[2mm] x_l(t)^\top S_{BC} x_j \cdot \Delta_d(x_j(t)) \cdot \exp\left(-a_d(t)\sum_{k=j+1}^{l} \Delta_d(x_k(t))\right), & \text{if } l > j; \end{cases} \tag{16}$$

- the step size function at the $d$-th channel $\Delta_d \colon \mathbb{R}^D \to \mathbb{R}$ is defined by

$$\Delta_d(u) = \text{softplus}(S_{\Delta,d}(t) \cdot u) = \ln\left(1 + e^{S_{\Delta,d}\cdot u}\right), \qquad u \in \mathbb{R}^D. \tag{17}$$

The dynamical system (6) is parameterized by the input-output matrix

$$S_{BC} = S_C^\top S_B \in \mathbb{R}^{D \times D},$$

the step-size matrix

$$S_\Delta = [S_{\Delta,1}, \ldots, S_{\Delta,D}] \in \mathbb{R}^{D \times D},$$

and the state coefficient vector

$$a = (a_1, \ldots, a_D) \in \mathbb{R}^D_{>0}.$$

Our main problem is studying the dynamical properties of the solutions of system (6). In particular, we want to understand how the set $\lim_{t \to +\infty}\{x_1(t), \ldots, x_L(t)\}$ looks like and when it reduces to a set of a single point.

**Notation.** Throughout this section, the notation $O(1)$ denotes a universal positive constant. We write $A \lesssim B$ to mean that there exists a constant $C_0 > 0$, independent of time, such that $|A| \leq C_0|B|$.

## B.1  POSSIBLE LIMIT POINTS OF THE FIRST TOKEN

In this subsection, we will characterize all possible limit points for the first token. In order to do that, let us consider the differential equation governing the trajectory of the first equation, which is the first equation in the dynamical system (14), given by

$$\frac{d}{dt}x_1 = \mathbf{P}_{x_1}\big(A_{11}(x)x_1\big) \tag{18}$$

where

$$A_{11}(x) = \mathrm{diag}(A_{1,11}, A_{2,11}, \cdots, A_{d,11})$$

and $A_{d,11} = \langle x_1, S_{BC}x_1 \rangle \Delta_d(x_1)$. We will determine the set of equilibrium points of this differential equation in Subsection B.1.1. However not every equilibrium point is a limit point. The set of possible actual limit points of the first token is then presented in Subsection B.1.2. Moreover, since

$$\langle S_{BC}x_1, x_1 \rangle = \langle Sx_1, x_1 \rangle,$$

where

$$S = \tfrac{1}{2}\big(S_{BC} + S_{BC}^\top\big)$$

denotes the symmetric part of $S_{BC}$, it follows that only the symmetric component contributes to the quadratic form. Throughout this section, we may therefore assume without loss of generality that $S_{BC}$ is symmetric.

### B.1.1  EQUILIBRIUM POINTS ACCORDING TO THE FIRST TOKEN

**Theorem B.1** (Equilibria of the first token). *A vector $y \in \mathbb{S}^{D-1}$ is an equilibrium of equation (18) if and only if it belongs to one of the following types:*

**Type 1.** *(Alignment equilibria) The vector $y$ may have some zero coordinates, but restricted to its nonzero entries we obtain*

$$y^{(\neq 0)} = (y_{k_1}, \ldots, y_{k_\ell}) \in \mathbb{S}^{\ell-1},$$

*which lies in the common kernel of the column differences*

$$S_{\Delta,k_j} - S_{\Delta,k_1}, \qquad j = 1, \ldots, \ell,$$

*for the active indices $\{k_1, \ldots, k_\ell\}$.*

**Type 2.** *(Quadratic constraint equilibria) The vector $y$ satisfies*

$$y^\top S_{BC}\, y = 0.$$

*If $S_{BC}$ is definite, then Type 2 equilibria do not exist. Furthermore, if the columns of $S_\Delta$ are linearly independent, then the Type 1 equilibria collapse to exactly two antipodal points*

$$y^{(\neq 0)} = \pm \xi,$$

*where $\xi$ is the unique unit vector in the one-dimensional intersection of the kernels above.*

*Proof.* By definition,

$$A_{11}(x) = \langle x_1, S_{BC}x_1 \rangle \, \mathrm{diag}\big(\Delta_1(x_1), \Delta_2(x_1), \ldots, \Delta_D(x_1)\big).$$

Substituting $x_1 = y = (y_1, \ldots, y_D) \in \mathbb{S}^{D-1}$, the dynamics take the form

$$y' = \mathbf{P}_y\big(A_{11}(y)y\big) = \langle S_{BC}y, y \rangle \, \mathbf{P}_y\big(\Delta_1(y)y_1, \ldots, \Delta_D(y)y_D\big),$$

where $\mathbf{P}_y$ denotes the orthogonal projection onto $T_y\mathbb{S}^{D-1}$.

For each component this yields

$$y_i' = \langle S_{BC} y, y \rangle \{ \Delta_i(y) - \Delta(y) \} y_i, \tag{19}$$

with

$$\Delta(y) = \sum_{k=1}^{D} \Delta_k(y) y_k^2, \qquad \Delta_i(y) = \ln\!\big(1 + e^{S_{\Delta,i} \cdot y}\big). \tag{20}$$

Equation (19) shows that equilibria arise in two distinct cases:

   (i) either $\langle S_{BC} y, y \rangle = 0$, corresponding to Type 2 equilibria,

   (ii) or else $\Delta_i(y) = \Delta(y)$ for every index $i$ with $y_i \neq 0$.

Since $|y| = 1$, there exists at least one index $i_0$ with $y_{i_0} \neq 0$. For any $i$ with $y_i \neq 0$, condition (ii) implies

$$\Delta_i(y) = \Delta_{i_0}(y),$$

which is equivalent to

$$(S_{\Delta,i} - S_{\Delta,i_0}) \cdot y = 0. \tag{21}$$

Let $\{k_1, \ldots, k_\ell\}$ denote the indices of the nonzero entries of $y$, and write

$$y^{(\neq 0)} = (y_{k_1}, \ldots, y_{k_\ell})$$

for the restriction of $y$ to these coordinates. Then (21) becomes

$$(S_{\Delta,k_j} - S_{\Delta,k_1}) \cdot y^{(\neq 0)} = 0, \qquad j = 1, \ldots, \ell,$$

which precisely characterizes Type 1 equilibria.

Finally, if $S_{BC}$ is definite, then $\langle S_{BC} y, y \rangle > 0$ for all $y \neq 0$, so no Type 2 equilibria can exist. Moreover, if the columns $\{S_{\Delta,k_j}\}_{j=1}^{\ell}$ are linearly independent, the intersection

$$\bigcap_{j=1}^{\ell} \langle S_{\Delta,k_j} - S_{\Delta,k_1} \rangle^{\perp}$$

is one-dimensional in $\mathbb{R}^{\ell}$. Its intersection with the unit sphere $\mathbb{S}^{\ell-1}$ therefore consists of exactly two antipodal points $\pm\xi$, which are the only possible equilibria. $\qquad\square$

### B.1.2   Limit points of the First Token

The analysis of stability near equilibrium points, in order to characterize the limit sets of equation (18), is in general quite delicate. Thanks to Theorem B.1, under the assumption that $S_{BC}$ is definite and the columns of $S_\Delta$ are linearly independent, the nonzero-part of equilibria are the two antipodal points

$$y^{(\neq 0)} = (y_{k_1}, \cdots, y_{k_\ell}) = \pm\xi,$$

where $\xi$ is the unique intersection of $\mathbb{S}^{\ell-1}$ with the one-dimensional subspace

$$\bigcap_{j=1}^{\ell} \ker\!\big((S_{\Delta,k_j} - S_{\Delta,k_1})^{\top}\big).$$

To avoid unnecessary technical complications, we shall henceforth restrict attention to this case. In particular, we consider the high-dimensional setting in which

$$S_{\Delta,m} = \pm e_m,$$

with $e_m$ denoting the $m$-th coordinate vector (see Theorem B.5). Apart from the special case $D = 2$, where the structure can be worked out explicitly (see Theorem B.7), this reduction allows us to describe all possible limit points of the first token.

We start with the following auxiliary lemma.

**Lemma B.2** (Sign Preservation). *Let $y(t) = (y_1(t), \ldots, y_D(t)) \in C^\infty([0, +\infty), \mathbb{R}^D)$ be a solution of equation* (18)*. If $y_i(0) < 0$ then $y_i(t) < 0$ for all $t > 0$. Similarly, if $y_i(0) > 0$ then $y_i(t) > 0$ for all $t > 0$.*

*Proof.* Suppose $y_i(0) < 0$ but $y_i(t) \geq 0$ for some $t > 0$. Then there exists a minimal $T > 0$ such that
$$y_i(T) = 0, \qquad y_i(t) < 0 \quad \text{for all } t \in [0, T).$$
By continuity of $y_i$, we have
$$\lim_{t \to T^-} y_i(t) = 0.$$

For $t \in (0, T)$ the evolution equation reads
$$\frac{d}{dt} y_i(t) = (S_{BC}y \cdot y)\big(\Delta_i(y) - \Delta(y)\big) y_i(t).$$

Dividing by $y_i(t)$ yields
$$\frac{d}{dt}\big(\ln |y_i(t)|\big) = (S_{BC}y \cdot y)\big(\Delta_i(y) - \Delta(y)\big).$$

Let $\{t_n\}_{n=1}^\infty \subset (0, T)$ be an increasing sequence with $t_n \to T$. Then
$$\ln |y_i(t_n)| - \ln |y_i(0)| = \int_0^{t_n} (S_{BC}y \cdot y)\big(\Delta_i(y) - \Delta(y)\big) ds.$$

The right-hand side is bounded, hence the left-hand side is also bounded. But since $|y_i(t_n)| \to 0$, we have $\ln |y_i(t_n)| \to -\infty$, a contradiction. Thus $y_i(t) < 0$ for all $t > 0$.

The case $y_i(0) > 0$ is analogous. $\qquad\square$

**Lemma B.3.** *Let $y(t) = (y_1(t), \ldots, y_D(t)) \in C^\infty([0, +\infty), \mathbb{R}^D)$ be a solution of equation* (18)*. Assume that $S_{\Delta,m} = \pm e_m$ and $y^\top S_{BC}y > 0$ for all $y \neq 0$. Here, the sign $\pm$ in $\pm e_m$ is the sign of $y_m(0)$. Then the following statements hold:*

    *(i)  If $0 < y_j(0) < y_i(0)$, then $0 < y_j(t) < y_i(t)$ for all $t > 0$.*

    *(ii)  If $y_i(0) < y_j(0) < 0$, then $y_i(t) < y_j(t) < 0$ for all $t > 0$.*

*Proof.* First, we claim that for all $t \geq 0$,
$$\Delta_i(y(t)) = \ln\big(1 + e^{|y_i(t)|}\big). \tag{22}$$

Indeed, if $y_i(0) < 0$, then by Lemma B.2 we have $y_i(t) < 0$ for all $t \geq 0$. Since $S_{\Delta,i} = -e_i$ by definition, it follows that
$$\Delta_i(y(t)) = \ln\big(1 + e^{-e_i \cdot y(t)}\big) = \ln\big(1 + e^{|y_i(t)|}\big).$$

This establishes (22); the case $y_i(0) > 0$ is analogous.

    (i)  First, let us consider the case when $0 < y_j(0) < y_i(0)$. According to equation (19) and (20), for $y_i > 0$, we have:
$$\frac{d}{dt}(\ln y_i) = S_{BC}y \cdot y\,(\Delta_i(y) - \Delta(y)).$$

       Hence
$$\frac{d}{dt}\ln\left(\frac{y_i}{y_j}\right) = S_{BC}y \cdot y\,(\Delta_i(y) - \Delta_j(y)).$$

    Let $c_{ij}(t) = \dfrac{y_i(t)}{y_j(t)}$. We have $c_{ij}(0) > 1$. We will show that $c_{ij}(t) > 1$ for all $t > 0$.

    Assume this is not true, then there is a smallest $T > 0$ such that $c_{ij}(T) = 1$ and $c_{ij}(t) > 1$ for all $t \in [0, T)$. By definition of $c_{ij}$, we have
$$y_i(t) > y_j(t) \qquad \forall t \in [0, T).$$

We have
$$\Delta_i(y) = \ln(1 + e^{y_i}),$$
which is a function in $y_i$. The function $y_i \mapsto \Delta_i(y)$ is increasing, hence on $[0, T)$ we have
$$\Delta_i(y) - \Delta_j(y) > 0.$$
Hence on $[0, T)$, we obtain
$$\frac{d}{dt} \ln(c_{ij}(t)) = (Sy \cdot y)(\Delta_i(y) - \Delta_j(y)) > 0,$$
so $c_{ij}(t)$ is increasing on $[0, T)$ with $c_{ij}(0) > 1$. Hence we cannot have $c_{ij}(T) = 1$. The proof of this case is complete.

$(ii)$ Next, let us consider the case when $y_i(0) < y_j(0) < 0$. In this case, we have
$$\Delta_i(y) = \ln(1 + e^{-y_i}), \quad \text{and} \quad \Delta_j(y) = \ln(1 + e^{-y_j}).$$
We have
$$\frac{d}{dt} \ln(c_{ij}(t)) = (Sy \cdot y)(\Delta_i(y) - \Delta_j(y)),$$
where $c_{ij}(t) = \dfrac{y_i(t)}{y_j(t)} > 0$. We see that $-y_i(0) > -y_j(0) > 0$, hence $c_{ij}(0) > 1$. We will show that $c_{ij}(t) > 1$ for all $t > 0$. Assume this is not true, then there is a smallest $T > 0$ such that $c_{ij}(T) = 1$ and $c_{ij}(t) > 1$ for all $t \in [0, T)$. By definition of $c_{ij}$, we have
$$-y_i(t) > -y_j(t) > 0 \qquad \forall t \in [0, T).$$
Hence
$$\Delta_i(y) > \Delta_j(y) \qquad \forall t \in [0, T),$$
which implies that
$$\frac{d}{dt} \ln(c_{ij}(t)) > 0.$$
Therefore $c_{ij}(t)$ is strictly increasing on $[0, T)$ with $c_{ij}(0) > 1$, making $c_{ij}(T) = 1$ impossible. Thus, $c_{ij}(t) > 1$ for all $t > 0$.

The proof is complete. $\qquad\square$

**Lemma B.4.** *Let $y(t) = (y_1(t), \ldots, y_D(t)) \in C^\infty([0, +\infty), \mathbb{R}^D)$ be a solution of equation* (18). *Assume that the symmetric part*
$$S = \tfrac{1}{2}(S_{BC} + S_{BC}^\top)$$
*of $S_{BC}$ is positive definite, and that $S_{\Delta,m} = \pm e_m$, where the sign $\pm$ agrees with the sign of $y_m(0)$. Then the following hold:*

$(i)$ *If $0 < y_j(0) < y_i(0)$, then either*
$$\lim_{t \to \infty} y_j(t) = 0 \quad or \quad \lim_{t \to \infty} \frac{y_i(t)}{y_j(t)} = 1.$$

$(ii)$ *If $y_i(0) < y_j(0) < 0$, then either*
$$\lim_{t \to \infty} y_j(t) = 0 \quad or \quad \lim_{t \to \infty} \frac{y_i(t)}{y_j(t)} = 1.$$

*Proof.* We prove case $(i)$; case $(ii)$ follows by the same argument.

Define
$$c(t) = \frac{y_i(t)}{y_j(t)}.$$
By Lemma B.3, we have $c(t) > 1$ for all $t \geq 0$. Differentiating and using (22), we obtain
$$\frac{d}{dt}\big(\ln c(t)\big) = \langle Sy, y \rangle \big(\ln(1 + e^{y_i}) - \ln(1 + e^{y_j})\big).$$

Let $f(t) = \ln(1 + e^t)$. By the mean value theorem, there exists $u \in (y_j, y_i)$ such that

$$f(y_i) - f(y_j) = f'(u)\,(y_i - y_j) = \frac{e^u}{1 + e^u}\,(y_i - y_j).$$

Since $\frac{1}{2} \leq \frac{e^u}{1+e^u} \leq 1$ and $y_i - y_j > 0$, it follows that

$$\tfrac{1}{2}\,\langle Sy, y\rangle\,(y_i - y_j) \;\leq\; \frac{d}{dt}\ln c(t) \;\leq\; \langle Sy, y\rangle\,(y_i - y_j).$$

Let $\lambda_m$ and $\lambda_M$ be the smallest and largest eigenvalues of $S$, respectively. Since $y \in \mathbb{S}^{D-1}$, we have

$$\lambda_m \;\leq\; \langle Sy, y\rangle \;\leq\; \lambda_M.$$

Hence,

$$\tfrac{1}{2}\lambda_m\,(y_i - y_j) \;\leq\; \frac{d}{dt}\ln c(t) \;\leq\; \lambda_M\,(y_i - y_j).$$

Equivalently,

$$\tfrac{1}{2}\lambda_m\,y_j(t)\,(c^2(t) - c(t)) \;\leq\; c'(t) \;\leq\; \lambda_M\,y_j(t)\,(c^2(t) - c(t)).$$

Since $c(0) > 1$, the comparison principle gives

$$1 < c(t) \leq z(t), \qquad t \geq 0, \tag{23}$$

where $z$ solves

$$z'(t) = \lambda_M\,y_j(t)\,(z^2 - z), \qquad z(0) = c(0).$$

The explicit solution is

$$z(t) = \frac{\frac{c(0)}{c(0)-1}\,\exp\!\left(\lambda_M \int_0^t y_j(s)\,ds\right)}{\frac{c(0)}{c(0)-1}\,\exp\!\left(\lambda_M \int_0^t y_j(s)\,ds\right) - 1}.$$

Therefore, the inequality (23) is equivalent to

$$\frac{c(0)}{c(0) - 1}\,\exp\!\left(\lambda_M \int_0^t y_j(s)\,ds\right) \;\leq\; \frac{c(t)}{c(t) - 1}, \qquad t > 0.$$

If $\int_0^\infty y_j(s)\,ds = \infty$, then the left-hand side diverges to $+\infty$, and hence

$$\lim_{t \to \infty} \frac{c(t)}{c(t) - 1} = \infty,$$

which implies

$$\lim_{t \to \infty} \frac{c(t) - 1}{c(t)} = 0.$$

Thus,

$$\lim_{t \to \infty} c(t) = \lim_{t \to \infty} \frac{y_i(t)}{y_j(t)} = 1.$$

If instead $\int_0^\infty y_j(s)\,ds < \infty$, then necessarily

$$\lim_{t \to \infty} y_j(t) = 0.$$

This completes the proof. $\qquad\square$

**Theorem B.5** (Convergence of the first tokens in high dimension). *Assume that the symmetric part of $S_{BC}$ is positive definite. Let $y(t) = (y_1(t), \dots, y_D(t)) \in C^\infty([0, \infty), \mathbb{R}^D)$ be a solution of equation (18). Assume that $S_{\Delta,m} = \pm e_m$, where the sign is chosen according to $\mathrm{sgn}(y_m(0)) \neq 0$, for all $m = 1, \dots, D$. Then*

$$\lim_{t \to \infty} y(t) = \pm e_d,$$

*where $d$ is the index of the largest initial coordinate in absolute value, i.e.*

$$|y_d(0)| = \max_{1 \leq i \leq D} |y_i(0)|,$$

*and the sign is $\mathrm{sgn}(y_d(0))$.*

*Proof.* Without loss of generality, we may assume that $d = D$, i.e.

$$|y_D(0)| = \max_{1 \leq i \leq D} |y_i(0)|.$$

By reordering the coordinates if necessary, we may further assume that

$$y_D(0) = \min\{\, y_i(0) : y_i(0) < 0 \,\} \quad \text{or} \quad y_D(0) = \max\{\, y_i(0) : y_i(0) > 0 \,\}. \tag{24}$$

By Lemma B.2, the ordering of $\{y_i(t)\}_{i=1}^D$ (from smallest to largest) is preserved for all $t \geq 0$. Moreover, by Lemma B.4, any limit point $\xi$ of $y(t)$ must be of the form

$$\xi = (-c, \ldots, -c,\, 0, \ldots, 0,\, p, \ldots, p)$$

for some constants $c, p > 0$.

For such a vector $\xi$, we compute

$$\Delta_1(\xi) = \ln(1 + e^c), \qquad \Delta_D(\xi) = \ln(1 + e^p).$$

Hence $c = p$, so

$$\xi = (-p, \ldots, -p,\, 0, \ldots, 0,\, p, \ldots, p).$$

For $\xi$ to be an equilibrium, it must satisfy

$$\Delta_D(\xi) = \Delta(\xi).$$

This condition yields

$$\ln(1 + e^p) = \sum_{i=1}^D \ln(1 + e^p)\, \xi_i^2 = \ln(1 + e^p) \sum_{\xi_i \neq 0} \xi_i^2 = |\{i : \xi_i = \pm p\}|\, \ln(1 + e^p).$$

Thus the set $\{i : \xi_i = \pm p\}$ must contain exactly one index.

Consequently, there are only two possible equilibria:

$$\xi = (-p, \ldots, 0, 0) \quad \text{or} \quad \xi = (0, \ldots, 0, p).$$

These two cases correspond precisely to the alternatives in (24). This completes the proof.

$\square$

**Remark B.6.** The conclusion of Theorem B.5 highlights a striking selection mechanism: under the assumption that the columns of $S_\Delta$ are linearly independent, the long-time dynamics do not converge to a large collection of equilibria lying in a lower-dimensional sphere in Theorem B.1. Instead, the trajectory is forced to pick out one of the most "extreme" coordinate directions, namely the basis vector $\pm e_d$ corresponding to the largest initial coordinate in magnitude.

In the two-dimensional case, we can actually determine all possible limit points of the first token in the following theorem.

**Theorem B.7** (Convergence of the first token in two dimensions)**.** *Let $D = 2$ and let $y(t) = (y_1(t), y_2(t)) \in C^\infty([0, +\infty), \mathbb{R}^2)$ be a solution of* (18)*. Assume that*

- *$S_{BC} = \mathrm{diag}(\lambda_1, \lambda_2)$ with either $\lambda_1 \neq \lambda_2$ or $\lambda_1 = \lambda_2 \neq 0$;*

- *$S_{\Delta,2} \neq S_{\Delta,1}$.*

*In this setting, the unit circle $\mathbb{S}^1$ intersects the line $\langle S_{\Delta,2} - S_{\Delta,1} \rangle^\perp$ at two antipodal points $\pm\xi$. Then the limit $\lim_{t \to +\infty} y(t)$ exists and belongs to the set*

$$\Omega = \{(0, \pm 1), (\pm 1, 0)\} \cup \{\pm\xi\} \cup \left\{ (\cos(\varphi_\star), \sin(\varphi_\star)) : \cos(2\varphi_\star) = \tfrac{\lambda_1 + \lambda_2}{\lambda_1 - \lambda_2} \right\}.$$

*Proof.* Since $D = 2$, we may parametrize

$$y(t) = (\cos\varphi(t),\, \sin\varphi(t)), \qquad \varphi(t) \in [0, 2\pi].$$

Define $y^\perp = (-y_2, y_1)$. Then

$$\varphi'(t)y^\perp = y'(t) = A_{11}y - (A_{11}y \cdot y)\,y.$$

Taking the inner product with $y^\perp$, and using $|y| = 1$ and $y \cdot y^\perp = 0$, yields

$$\varphi'(t) = A_{11}y \cdot y^\perp = (\lambda_1 y_1^2 + \lambda_2 y_2^2)y_1 y_2\big(\Delta_2(y) - \Delta_1(y)\big).$$

Substituting $y = (\cos\varphi, \sin\varphi)$, we obtain

$$\varphi'(t) = \big[(\lambda_1 + \lambda_2) + (\lambda_1 - \lambda_2)\cos(2\varphi)\big]\sin(2\varphi)\big[h(R_2, \varphi - \beta) - h(R_1, \varphi - \alpha)\big],$$

where

$$h(R, u) = \tfrac{1}{4}\ln\big(1 + e^{R\cos u}\big), \qquad S_{\Delta,2} = R_2(\cos\beta, \sin\beta), \qquad S_{\Delta,1} = R_1(\cos\alpha, \sin\alpha).$$

Thus $\varphi$ satisfies the autonomous ODE

$$\varphi' = F(\varphi),$$

where the set of zeros of $F$ in $[0, 2\pi]$ is precisely $\Omega$. Since $\Omega$ is finite, we may partition $(0, 2\pi)$ into disjoint intervals with endpoints in $\Omega$ such that $F$ has a definite sign on each interval. By the phase portrait of $\varphi$, every trajectory converges to one of the equilibria of $F$, i.e. to a point in $\Omega$. This proves the claim. $\qquad\square$

### B.2 ALL OTHER TOKENS ARE ATTRACTED BY THE FIRST TOKEN

Different from the first token, the study of the convergence of all tokens based on the dynamical system (14) is much more challenging. To make the proofs feasible while maintaining the generality of our theoretical findings, in this whole subsection, we will consider the dynamical system (14) under the following assumptions:

*Assumption:* $S_{\Delta,1} = \ldots = S_{\Delta,D} =: \alpha \in \mathbb{R}^D$ and $a_1 = \ldots = a_D =: a \in \mathbb{R}$.

In this case we can rewrite dynamical system (14) as:

$$\frac{d}{dt}x_i = \sum_{j=1}^{i-1} A_{ij}\mathbf{P}_{x_i}(x_j) \tag{25}$$

where $A_{ij}$ is defined as

$$A_{ij} = (x_i \cdot S_{BC}x_j)a_{ij}, \qquad a_{ij} = \Delta(x_i)e^{-a\sum_{k=j+1}^{i}\Delta(x_k)}, \qquad \text{and} \qquad \Delta(x_i) = \ln(1 + e^{\alpha \cdot x_i}).$$

Throughout this section, we will assume that $x(t) = (x_1(t), x_2(t), \ldots, x_L(t)) \in C^\infty([0, +\infty); \mathbb{R}^D)^L$ is a solution of the dynamical system (25).

For the first token $x_1(t)$, we see that

$$\frac{d}{dt}x_1 = A_{11}\mathbf{P}_{x_1}(x_1) = 0$$

This implies that $x_1(t) = x_1$ is a constant for all $t \geq 0$. In the following, we will present all possible limit points of $x_l(t)$ with $l = 2, \ldots, L$.

Define the consensus set

$$\Omega_0 = \{(x_1, \pm x_1, \ldots, \pm x_1)(0) \in (\mathbb{S}^{D-1})^L\}.$$

This set has exactly $2^{L-1}$ distinct elements, corresponding to the situation where every token is either aligned or anti-aligned with $x_1$.

Our main theorem of this section is stated as follows:

**Theorem B.8.** *Consider $L$ trajectories $x_1(t), \ldots, x_L(t)$ evolving on the unit sphere $\mathbb{S}^{D-1}$ under the dynamics* (25). *Assume that $S_{BC}$ is a symmetric continuous map (possibly nonlinear) on $\mathbb{S}^{D-1}$, and that the initial reference token $x_1 = x_1(0)$ satisfies*

$$\langle x_1, S_{BC}x_1 \rangle > 0.$$

*Suppose further that for each $i \geq 2$,*

$$\langle x_i(0), S_{BC}x_1 \rangle \notin \{0, -1, 1\}, \qquad \langle x_i(0), x_1 \rangle \notin \{-1, 1\}.$$

*Then there exists a compact set $\mathcal{K} \subset (\mathbb{S}^{D-1})^L \setminus \Omega_0$, depending only on $x_1$ and $S_{BC}x_1$, such that for any initialization $(x_1, \ldots, x_L)(0)$, precisely one of the following holds:*

*(i) Consensus. The trajectory $x(t) = (x_1, \ldots, x_L)(t)$ converges to some $\omega_0 \in \Omega_0$ as $t \to \infty$. Moreover, there exists $C_0 > 0$ such that*

$$\|x(t) - \omega_0\| \lesssim e^{-C_0 t}, \qquad t \geq 0.$$

*(ii) Non-consensus. The trajectory admits a limit point in $\mathcal{K}$ as $t \to \infty$. For any $\omega \in \mathcal{K}$, the set of initial conditions whose trajectories admit $\omega$ as a limit point has measure zero in $(\mathbb{S}^{D-1})^L$.*

**Outline of the proof.** The structure of this section is as follows. We first establish an affirmative result: under a favorable positive-angle condition on the initial token configuration, the system converges to a state where all tokens align with $x_1$. This is proved in Proposition B.9, with the base case $L = 2$ treated separately in Proposition B.15. We then proceed by induction: assuming exponential convergence of the first $k - 1$ tokens to $\pm x_1$, we show that the $k$-th token either converges exponentially to $\pm x_1$ or to a point in the orthogonal complement $\langle S_{BC}x_1 \rangle^{\perp}$. Finally, Proposition B.13 demonstrates that the latter scenario occurs only for a set of initial conditions of measure zero. The main difficulty arises from the fact that the interaction terms $\langle x_i, S_{BC}x_j \rangle$ are not sign-definite—a feature that distinguishes the model from classical consensus or synchronization dynamics. This necessitates a careful combination of nonlinear analysis and linear stability techniques to exclude nongeneric limiting behaviors.

**Proposition B.9.** *Assume that $\langle x_1(0), S_{BC}x_1(0) \rangle > 0$ and that:*

- $\langle x_i(0), S_{BC}x_j(0) \rangle > 0$ *for all $i, j \in [1, L]$, and*

- $\langle x_i(0), x_1(0) \rangle \notin \{-1, 1\}$ *for all $i \in [2, L]$.*

*Then,*

$$\sum_{i=1}^{L} |x_i(t) - x_1(0)| \lesssim e^{-O(1)t}.$$

*Moreover, we have*

$$\langle x_i(t), S_{BC}x_j(t) \rangle > 0 \quad \text{for all } t > 0.$$

The proof of Proposition B.9 will be postponed to the end of this section, as it requires several preparatory steps. To analyze the token dynamics, we first encode the pairwise interactions between tokens through the scalar quantities

$$X_{ij}(t) = \langle x_i(t), x_j(t) \rangle, \qquad U_{ij}(t) = \langle x_i(t), S_{BC}x_j(t) \rangle, \qquad i, j \in [1, L].$$

Here $X_{ij}$ measures the relative alignment between tokens $i$ and $j$ on the sphere $\mathbb{S}^{d-1}$, while $U_{ij}$ captures their interaction through the transformation $S_{BC}$. These variables will serve as the fundamental building blocks in our nonlinear analysis and will play a central role in the proof of Theorem B.8.

Our first step is to establish their evolution equations.

**Lemma B.10.** *Assume that $S_{BC}$ is symmetric. For $i > 1$, we have*

$$X'_{i1}(t) = \sum_{j=1}^{i-1} a_{ij} U_{ij} \left[ X_{j1} - X_{ij} X_{i1} \right],$$

$$U'_{i1}(t) = \sum_{j=1}^{i-1} a_{ij} U_{ij} \left[ U_{j1} - X_{ij} U_{i1} \right]. \tag{26}$$

*If $i \geq j > 1$, then*

$$X'_{ij}(t) = \sum_{k=1}^{i-1} a_{ik} U_{ik} \left[ X_{kj} - X_{ik} X_{ij} \right] + \sum_{k=1}^{j-1} a_{jk} U_{jk} \left[ X_{ki} - X_{jk} X_{ji} \right],$$

$$U'_{ij}(t) = \sum_{k=1}^{i-1} a_{ik} U_{ik} \left[ U_{kj} - X_{ik} U_{ij} \right] + \sum_{k=1}^{j-1} a_{jk} U_{jk} \left[ U_{ki} - X_{jk} U_{ji} \right].$$

*Proof.* Recalling (25), we have

$$\frac{d}{dt}x_i = \sum_{j=1}^{i-1} a_{ij} U_{ij} \left[ x_j - X_{ij} x_i \right].$$

Dotting both sides of the above equation with $x_1$, we have

$$X'_{i1}(t) = \sum_{j=1}^{i-1} a_{ij} U_{ij} \left[ X_{j1} - X_{ij} X_{i1} \right].$$

Next for $i \geq j > 1$, we compute

$$\begin{aligned}
X'_{ij} &= \frac{d}{dt} \langle x_i(t), x_j(t) \rangle = \frac{d}{dt} x_i \cdot x_j + \frac{d}{dt} x_j \cdot x_i \\
&= \sum_{k=1}^{i-1} a_{ik} U_{ik} (x_k \cdot x_j - X_{ik} x_i \cdot x_j) + \sum_{k=1}^{j-1} a_{jk} U_{jk} (x_k \cdot x_i - X_{jk} x_j \cdot x_i) \\
&= \sum_{k=1}^{i-1} a_{ik} U_{ik} (X_{kj} - X_{ik} X_{ij}) + \sum_{k=1}^{j-1} a_{jk} U_{jk} (X_{ki} - X_{jk} X_{ji}).
\end{aligned}$$

We also have

$$\frac{d}{dt} U_{i1} = \frac{d}{dt} x_i \cdot S_{BC} x_1 = \sum_{j=1}^{i-1} a_{ij} U_{ij} \left[ U_{j1} - X_{ij} U_{i1} \right],$$

and similarly

$$\begin{aligned}
\frac{d}{dt} U_{ij} &= \frac{d}{dt} x_i \cdot S_{BC} x_j + \frac{d}{dt} x_j \cdot S_{BC} x_i \\
&= \sum_{k=1}^{i-1} a_{ik} U_{ik} \left[ U_{kj} - X_{ik} U_{ij} \right] + \sum_{k=1}^{j-1} a_{jk} U_{jk} \left[ U_{ki} - X_{jk} U_{ji} \right].
\end{aligned}$$

The proof is complete. $\qquad\square$

**Proposition B.11.** *Assume that $S_{BC}$ is symmetric and that $x_1 = x_1(0)$ satisfies*

$$\langle x_i(0), S_{BC} x_1 \rangle \notin \{-1, 0, 1\}, \qquad \textit{for all } i \in [2, L].$$

*Then there exists a minimal integer $k \in [3, L]$ such that exactly one of the following holds:*

*(i) For every $i \leq k$, we have $x_i(t) \to \pm x_1(0)$ as $t \to \infty$, with exponential convergence rate*

$$\sum_{i=1}^{k} \left| x_i(t) - (\pm x_1(0)) \right| \lesssim e^{-O(1)t}.$$

*(ii) For every $i \leq k - 1$, we have $x_i(t) \to \pm x_1(0)$ as $t \to \infty$, while*

$$\lim_{t \to \infty} \langle x_k(t), S_{BC} x_1(0) \rangle = 0,$$

*with exponential decay*

$$\left| \langle x_k(t), S_{BC} x_1(0) \rangle \right| \lesssim e^{-O(1)t}.$$

*Moreover, after possibly replacing each $x_i$ by its reflection $\widetilde{x}_i = -x_i$, we may assume without loss of generality that*

$$\lim_{t \to \infty} \widetilde{x}_i(t) = x_1(0), \qquad \textit{for all } i \leq k - 1.$$

*Proof.* For simplicity of notation, we may assume $k = L$ and proceed by induction on $L$. Our main tool is Proposition B.9.

We first consider the case $L = 2$. If $\langle x_2(0), S_{BC}x_1 \rangle > 0$, then the result follows directly from Proposition B.9. If instead $\langle x_2(0), S_{BC}x_1 \rangle < 0$, we define

$$\widetilde{x}_2(t) = -x_2(t).$$

Since $\mathbf{P}_{-y}(x) = \mathbf{P}_y(x)$ for all $x, y \in \mathbb{S}^{d-1}$, we obtain

$$\frac{d}{dt}\widetilde{x}_2 = -\frac{d}{dt}x_2 = -a_{21}\langle x_2, S_{BC}x_1 \rangle \mathbf{P}_{x_2}(x_1) = a_{21}\langle \widetilde{x}_2, S_{BC}x_1 \rangle \mathbf{P}_{\widetilde{x}_2}(x_1).$$

Hence, by Proposition B.9 with $\langle \widetilde{x}_2(0), S_{BC}x_1 \rangle > 0$, we deduce

$$|\widetilde{x}_2(t) - x_1| \lesssim e^{-O(1)t}.$$

From this point onward, starting from the third token, we replace $(x_1, x_2)$ by $(x_1, \widetilde{x}_2)$. We call this a positive angle rearrangement. By abuse of notation, $\widetilde{x}_i$ may denote either $x_i$ or $-x_i$, depending on whether $x_i \to x_1$ or $x_i \to -x_1$. This completes the case $L = 2$.

Now assume we have the rearrangement $(x_1, \widetilde{x}_2, \cdots, \widetilde{x}_{L-1})$ such that

$$\sum_{i=2}^{L-1} |\widetilde{x}_i(t) - x_1| \lesssim e^{-O(1)t}.$$

We claim that: if $\langle x_L(t), S_{BC}x_1 \rangle \nrightarrow 0$ as $t \to \infty$, then $x_L(t) \to \pm x_1$ as $t \to \infty$. Indeed, since $\langle x_L(t), S_{BC}x_1 \rangle \nrightarrow 0$ as $t \to \infty$, there exists $\epsilon_0 > 0$ such that either the set

$$\{t > 0: \quad \langle x_L(t), S_{BC}x_1 \rangle \geq \epsilon_0\},$$

or

$$\{t > 0: \quad \langle x_L(t), S_{BC}x_1 \rangle \leq -\epsilon_0\}$$

contains an increasing sequence of times $\{t_n\}_{n=1}^{\infty}$. We will show that there exists $n$ such that

$$\langle x_L(t_n), S_{BC}x_1 \rangle, \cdots, \langle x_L(t_n), S_{BC}\widetilde{x}_{L-1}(t_n) \rangle$$

all have the same sign (positive or negative). If this is the case, we can perform the reflection $\widetilde{x}_L$, apply Proposition B.9, and conclude $\widetilde{x}_L \to x_1$ as $t \to \infty$.

Since $\widetilde{x}_k \to x_1$ for all $k \leq L - 1$ as $t \to \infty$, there exists $T > 0$ such that for any $k \in [2, L-1]$,

$$|\langle x_L(t), S_{BC}x_1 \rangle - \langle x_L(t), S_{BC}\widetilde{x}_k(t) \rangle| \leq 0.1\epsilon_0 \quad \text{for all} \quad t \geq T.$$

This implies

$$\langle x_L(t), S_{BC}x_1 \rangle - 0.1\epsilon_0 \leq \langle x_L(t), S_{BC}x_k(t) \rangle \leq \langle x_L(t), S_{BC}x_1 \rangle + 0.1\epsilon_0 \quad \forall t \geq T.$$

- If there is an increasing sequence of times $\{t_n\}$ such that $\langle x_L(t_n), S_{BC}x_1 \rangle \geq \epsilon_0$, then we can find $t_n \geq T$ such that

$$\langle x_L(t_n), S_{BC}x_k(t_n) \rangle \geq 0.9\epsilon_0.$$

  Hence Proposition B.9 applied to $(x_1, \widetilde{x}_2, \cdots, x_L)$ at time $t = t_n$ yields $x_L(t) \to x_1$ as $t \to \infty$.

- If there is an increasing sequence of times $\{t_n\}$ such that $\langle x_L(t_n), S_{BC}x_1 \rangle \leq -\epsilon_0$, then we can find $t_n \geq T$ such that

$$\langle x_L(t_n), S_{BC}x_k(t_n) \rangle \leq -0.9\epsilon_0.$$

  Hence Proposition B.9 applied to $(x_1, \widetilde{x}_2, \cdots, \widetilde{x}_L)$ at time $t = t_n$ yields $\widetilde{x}_L(t) \to x_1$ as $t \to \infty$.

The claim is proved.

Now if $\langle x_L(t), S_{BC}x_1 \rangle \to 0$, we show that

$$|\langle x_L(t), S_{BC}x_1 \rangle| \lesssim e^{-O(1)t}.$$

Recalling that $U_{L1}(t) = \langle x_L(t), S_{BC}x_1 \rangle$, the above is equivalent to

$$U_{L1}^2(t) \to 0 \quad \text{as} \quad t \to \infty.$$

Using the evolution equation of $U_{L1}$ in B.10 and the fact that all previous tokens converge to $\pm x_1(0)$, there exists a time $T > 0$ and positive constants $c_T, \sigma, C_\sigma > 0$ such that

$$\frac{d}{dt}U_{L1}^2 \geq c_T U_{L1}^2 - C_\sigma e^{-\sigma t} \quad \text{for all} \quad t \geq T. \tag{27}$$

Hence for any time $T_1 \geq T$, we obtain

$$U_{L1}^2(t) \geq e^{c_0(t-T_1)}\left[U(T_1)^2 - \frac{C_\sigma}{c_T + \sigma}e^{-\sigma T_1}\right] + \frac{C_\sigma}{c_T + \sigma}e^{-\sigma t}, \qquad t \geq T_1.$$

Since $U_{L1}^2(t) \in [0,1]$ for all $t \geq 0$, by letting $t \to \infty$ it follows that

$$U(T_1)^2 \leq \frac{C_\sigma}{c_T + \sigma}e^{-\sigma T_1}.$$

The proof is complete. $\square$

**Remark B.12.** Proposition B.11 shows that, starting from the third token onward, each trajectory either converges to a point in $\{\pm x_1\}$ or else its component in the $S_{BC}x_1(0)$ direction vanishes asymptotically.

In the next proposition, we establish that if all preceding tokens converge to $\{\pm x_1\}$, then the probability that a subsequent token converges to any point outside $\{\pm x_1\}$ is zero.

**Proposition B.13.** *Let $L \geq 3$ and assume that $\langle S_{BC}x_1(0), x_1(0) \rangle > 0$. Fix any $\xi \in \mathbb{S}^{D-1}$ such that*

$$\langle \xi, S_{BC}x_1(0) \rangle = 0.$$

*Then the set of initial configurations $(x_2(0), \ldots, x_L(0)) \in (\mathbb{S}^{D-1})^{L-1}$ for which the corresponding trajectory satisfies*

$$(x_2(t), \ldots, x_{L-1}(t), x_L(t)) \longrightarrow (\pm x_1, \ldots, \pm x_1, \xi) \quad \text{as } t \to \infty$$

*has measure zero in $(\mathbb{S}^{D-1})^{L-1}$.*

*Proof.* Without loss of generality, we may assume that $x_k \to x_1$ for all $k \in [2, L-1]$. Our goal is to show that the set of initial conditions leading to $x_L \to \xi$ has measure zero. Recall the notation

$$X_{ij} = x_i \cdot x_j, \qquad U_{ij} = x_i \cdot S_{BC}x_j.$$

For convenience, we define the limiting configuration associated with $\xi$ by

$$x_\star = x_\star(\xi) := (x_1, \ldots, x_1, \xi) \in (\mathbb{S}^{D-1})^L.$$

We perturb the configuration $(x_i)_{i=1}^L$ around $x_\star$ by setting

$$\begin{cases} \widetilde{x}_1 = 0, \quad x_i = x_1 + \delta\widetilde{x}_i & \text{for } 2 \leq i \leq L-1, \\ x_L = \xi + \delta\widetilde{x}_L, \\ (x_1, \ldots, x_L) = x_\star + \delta\left(0, \widetilde{x}_2, \ldots, \widetilde{x}_L\right). \end{cases}$$

For $i, j \leq L$, we introduce the notation

$$\widetilde{U}_{ij} = \widetilde{x}_i \cdot S_{BC}\widetilde{x}_j,$$

and we set

$$\gamma = S_{BC}x_1 \cdot x_1 > 0.$$

We obtain the following first-order expansions:

$$
\begin{cases}
X_{Li} &= \xi \cdot x_1 + \delta\big[\widetilde{X}_{L1} + \xi \cdot \widetilde{x}_i\big] + O(\delta^2), \qquad 1 \le i \le L-1, \\[4pt]
U_{Li} &= \delta\big[\xi \cdot \widetilde{x}_i + \widetilde{U}_{L1}\big] + O(\delta^2), \qquad 1 \le i \le L-1, \\[4pt]
X_{ij} &= 1 + 2\delta\widetilde{X}_{ij} + O(\delta^2), \qquad 1 \le i,j \le L-1, \\[4pt]
U_{ij} &= S x_1 \cdot x_1 + \delta\, S_{BC} x_1 \cdot (\widetilde{x}_i + \widetilde{x}_j) + O(\delta^2), \qquad 1 \le i,j \le L-1.
\end{cases}
$$

Substituting the above expansions into the equations of Lemma B.10 and collecting the $O(\delta)$ terms, we obtain:

(i) Linearized equation for $U_{L1}$:

$$
\gamma^{-1}\, \partial_t \widetilde{U}_{L1} = a_{L1}(x_\star)\widetilde{U}_{L1} + \sum_{j=2}^{L-1} a_{Lj}(x_\star)\big[\widetilde{U}_{L1} + \xi \cdot \widetilde{x}_j\big].
$$

(ii) For $2 \le j \le L-1$, we have

$$
\gamma^{-1}\, \partial_t\big[\xi \cdot \widetilde{x}_j + \widetilde{U}_{L1}\big] = a_{L1}(x_\star)\widetilde{U}_{L1} + \Big(a_{Lj}(x_\star) - \sum_{k=1}^{j-1} a_{jk}(x_\star)\Big)\big[\xi \cdot \widetilde{x}_j + \widetilde{U}_{L1}\big]
$$

$$
+ \sum_{k=2}^{j-1} a_{jk}(x_\star)\big[\xi \cdot \widetilde{x}_k + \widetilde{U}_{L1}\big] + \sum_{\substack{2 \le k \le L-1 \\ k \ne j}} a_{Lk}(x_\star)\big[\xi \cdot \widetilde{x}_k + \widetilde{U}_{L1}\big].
$$

We now introduce the vector $\vec{X} \in \mathbb{R}^{L-1}$ defined by

$$
\vec{X}(t) = \begin{bmatrix} \widetilde{U}_{L1} \\ \xi \cdot \widetilde{x}_2 + \widetilde{U}_{L1} \\ \vdots \\ \xi \cdot \widetilde{x}_{L-1} + \widetilde{U}_{L1} \end{bmatrix} = [X_1, X_2, \ldots, X_{L-1}]^T.
$$

Formally, the vector $\vec{X}$ collects the perturbation coordinates that encode the angles formed by $(\widetilde{x}_2, \widetilde{x}_3, \ldots, \widetilde{x}_L)$ relative to the limiting configuration $x_\star \in (\mathbb{S}^{D-1})^L$. From the previous computations we obtain

$$
\gamma^{-1}\, \partial_t X_1 = a_{L1} X_1 + \sum_{j=2}^{L-1} a_{Lj} X_j,
$$

$$
\gamma^{-1}\, \partial_t X_j = a_{L1} X_1 + \sum_{k=2}^{j-1}(a_{jk} + a_{Lk})X_k + \Big(a_{Lj} - \sum_{k=1}^{j-1} a_{jk}\Big)X_j + \sum_{k=j+1}^{L-2} a_{Lk} X_k, \qquad 2 \le j \le L-1.
$$

Hence the vector $\vec{X}$ satisfies the linear system

$$
\gamma^{-1}\, \partial_t \vec{X}(t) = J(x_\star)\, \vec{X}(t),
$$

where $J(x_\star) \in \mathbb{R}^{(L-1)\times(L-1)}$ is the Jacobian matrix with rows $R_i = (J(x_\star))_{i,:}$ defined as follows (all coefficients $a_{ij}$ are evaluated at $x_\star = x_\star(\xi)$):

$$R_1 = \begin{bmatrix} a_{L1} & a_{L2} & a_{L3} & \cdots & a_{L,L-1} \end{bmatrix},$$

$$R_2 = \begin{bmatrix} a_{L1} & a_{L2} - a_{21} & a_{L3} & \cdots & a_{L,L-1} \end{bmatrix},$$

$$R_3 = \begin{bmatrix} a_{L1} & a_{32} + a_{L2} & a_{L3} - (a_{31} + a_{32}) & \cdots & a_{L,L-1} \end{bmatrix},$$

$$\vdots$$

$$R_{L-2} = \begin{bmatrix} a_{L1} & a_{L-2,2} + a_{L2} & a_{L-2,3} + a_{L3} & \cdots & a_{L-2,L-3} + a_{L,L-3} & a_{L,L-2} - \sum_{k=1}^{L-3} a_{L-2,k} & a_{L,L-1} \end{bmatrix},$$

$$R_{L-1} = \begin{bmatrix} a_{L1} & a_{L-1,2} + a_{L2} & \cdots & a_{L-1,L-2} + a_{L,L-2} & a_{L,L-1} - \sum_{k=1}^{L-2} a_{L-1,k} \end{bmatrix}.$$

We now show that the matrix $J(x_\star)$ always possesses at least one positive eigenvalue. To this end, subtract the first row $R_1$ from each of the rows $R_2, R_3, \ldots, R_{L-1}$. This row operation leaves the eigenvalues unchanged and yields an equivalent matrix with the following row structure:

$$\widetilde{R}_1 = \begin{bmatrix} a_{L1} & a_{L2} & a_{L3} & \cdots & a_{L,L-1} \end{bmatrix},$$

$$\widetilde{R}_2 = \begin{bmatrix} 0 & -a_{21} & 0 & \cdots & 0 \end{bmatrix},$$

$$\widetilde{R}_3 = \begin{bmatrix} 0 & a_{32} & -(a_{31} + a_{32}) & \cdots & 0 \end{bmatrix},$$

$$\vdots$$

$$\widetilde{R}_{L-2} = \begin{bmatrix} 0 & a_{L-2,2} & a_{L-2,3} & \cdots & a_{L-2,L-3} & -\sum_{k=1}^{L-3} a_{L-2,k} & 0 \end{bmatrix},$$

$$\widetilde{R}_{L-1} = \begin{bmatrix} 0 & a_{L-1,2} & \cdots & a_{L-1,L-2} & -\sum_{k=1}^{L-2} a_{L-1,k} \end{bmatrix}.$$

Denote the above matrix by $J_1$. Observe that $J_1^T$ has first row

$$\begin{bmatrix} a_{L1} & 0 & 0 & \cdots & 0 \end{bmatrix},$$

which shows that $a_{L1}(x_\star) > 0$ is an eigenvalue of $J_1$. Consequently, $J(x_\star)$ admits the positive eigenvalue $\lambda_+ = a_{L1}(x_\star) > 0$. We now prove that, for almost every initial configuration

$$\vec{x}_0 = (x_2(0), \ldots, x_L(0)) \in (\mathbb{S}^{D-1})^{L-1},$$

the trajectory converges with limits in $\{\pm x_1\}$. For $t \geq 0$, set

$$\vec{x}(t) = (x_2(t), x_3(t), \ldots, x_L(t)) \in (\mathbb{S}^{D-1})^{L-1}.$$

Define the flow map $\Phi : (\mathbb{S}^{D-1})^{L-1} \to (\mathbb{S}^{D-1})^{L-1}$ by

$$\vec{x}_0 = (x_2, \ldots, x_L)(0) \; \mapsto \; \vec{x}(t) = (x_2, \ldots, x_L)(t).$$

If $\vec{x}(t) = \Phi^t(\vec{x}_0)$ does not converge to $(x_1, \ldots, x_1)$ (up to a reflection of tokens), then by Proposition B.11 there exist an increasing sequence of rational times $\{t_n\}_{n=1}^\infty \subset \mathbb{Q}^+$ and some $\xi \in \langle S_{BC} x_1 \rangle^\perp$ (after a subsequence selection) such that

$$\vec{x}(t_n) = \Phi^{t_n}(\vec{x}_0) \; \to \; x_\star(\xi) \quad \text{as } n \to \infty.$$

Hence there exists $N \in \mathbb{N}$ such that for all $n \geq N$,

$$\Phi^{t_n}(\vec{x}_0) \in W_{\mathrm{loc}}^{\mathrm{sc}}(x_\star(\xi)), \tag{28}$$

where $W_{\mathrm{loc}}^{\mathrm{sc}}(x_\star(\xi))$ denotes the local center-stable manifold at $x_\star(\xi)$.

Since the linearization at $x_\star(\xi) \in (\mathbb{S}^{d-1})^{L-1}$ admits at least one positive eigenvalue,

$$\lambda_+(\xi) = a_{L1}(x_\star(\xi)) > 0,$$

in the direction of $x_\star(\xi)$, the local stable–center manifold $W_{\mathrm{loc}}^{\mathrm{sc}}(x_\star(\xi))$ has dimension at most $(d-1)(L-1) - 1$. Consequently, it has measure zero in $(\mathbb{S}^{d-1})^{L-1}$.

From (28) it follows that

$$\vec{x}_0 \in \bigcup_{t \in \mathbb{Q}^+} \Phi^{-t}(W^{\mathrm{sc}}_{\mathrm{loc}}(x_\star(\xi))),$$

and the right-hand side is a countable union of measure-zero sets. Therefore it also has measure zero. This proves the claim.

$\square$

Before proving Proposition B.9, we establish a useful lemma.

**Lemma B.14.** *Let $g : [t_0, \infty) \to \mathbb{R}$ be a differentiable function such that $g'$ is continuous. Assume $g(t_0) > \delta$ and*

$$g'(T) > 0 \quad \text{whenever } g(T) = \delta,$$

*then $g(t) \geq \delta$ for all $t \geq t_0$.*

*Proof.* Assume this is not true, since $g(t_0) > \delta$, there exists a smallest time $T > t_0$ such that $g(T) = \delta$. Since $g'$ is continuous, there exists $\epsilon_0 > 0$ such that

$$g'(t) > 0 \quad \text{for all} \quad t \in (T - \epsilon_0, T + \epsilon_0).$$

This implies $g(t) < g(T) = \delta$ for all $t \in (T - \epsilon_0, T)$. By continuity of $g$ and the fact that $g(t_0) > \delta$, there exists $T_1 \in (t_0, T)$ such that $g(T_1) = \delta$, violating the minimality of $T$. The proof is complete, and hence we have

$$g(t) > \delta \quad \text{for all} \quad t > t_0.$$

$\square$

The following establishes Proposition B.9 in the special case $L = 2$.

**Proposition B.15** (Proposition B.9 for $L = 2$)**.** *Assume that $\langle x_1(0), S_{BC} x_1(0) \rangle > 0$. Then*

$$\lim_{t \to \infty} x_2(t) = x_1(0) \quad \text{whenever} \quad \langle x_2(0), S_{BC} x_1(0) \rangle > 0.$$

*Moreover, the following hold:*

- *Exponential convergence:*

$$|x_2(t) - x_1(0)| \ \leq \ |x_2(0) - x_1(0)| \, e^{-O(1)\, t},$$

- *Positivity preservation:*

$$\langle x_2(t), S_{BC} x_1(0) \rangle > 0, \qquad \text{for all } t > 0.$$

*Proof.* Since

$$|x_2(t) - x_1|^2 = 2\big[1 - \langle x_1, x_2(t) \rangle\big],$$

it suffices to prove that

$$\lim_{t \to \infty} \langle x_2(t), x_1 \rangle = 1.$$

From the dynamical system (25), we have

$$\frac{d}{dt} x_2 = \langle x_2, S_{BC} x_1 \rangle \, a_{21} \big(x_1 - (x_1 \cdot x_2) x_2\big),$$

where $a_{21} > 0$ for all $t > 0$. Define

$$f(t) = \langle x_2(t), x_1 \rangle, \qquad g(t) = \langle x_2(t), S_{BC} x_1 \rangle.$$

A direct computation yields

$$f' = a_{21} g (1 - f^2),$$
$$g' = a_{21} g \, (x_1 \cdot S_{BC} x_1 - f g).$$

—

*Step 1: Positivity of $g(t)$.* We claim that

$$g(t) \ \geq \ \min\{ S_{BC}x_1 \cdot x_1, \ g(0)\}, \quad \text{for all} \quad t > 0. \tag{29}$$

At $t = 0$, we have $g(0) = \langle x_2(0), S_{BC}x_1 \rangle > 0$. Fix any $\delta \in (0, \min\{S_{BC}x_1 \cdot x_1, g(0)\})$. If $g(T) = \delta$ at some time $T$, then

$$g'(T) = a_{21}\delta\big(x_1 \cdot S_{BC}x_1 - \delta f(T)\big) \ \geq \ a_{21}\delta\big(S_{BC}x_1 \cdot x_1 - \delta\big) > 0.$$

By Lemma B.14, this implies $g(t) > \delta$ for all $t > 0$, proving (29).

—

*Step 2: Convergence of $f(t)$.* From (30),

$$f' = a_{21}g(1 - f^2),$$

with $a_{21}, g > 0$. Hence $f'(t) > 0$ whenever $f(t) \in (-1, 1)$, so $f$ is strictly increasing. Therefore, there exists $\theta = \sup_{t \geq 0} f(t) \in [-1, 1]$, with necessarily $\theta > -1$.

—

*Step 3: Exponential convergence.* For large $t$, we estimate

$$\frac{d}{dt}(1 - f) = -f' = -a_{21}g(1 - f)(1 + f) \ \lesssim \ -(1 + \theta)(1 - f(t)).$$

Integrating, we obtain

$$1 - f(t) \ \leq \ (1 - f(0))e^{-O(1)\,t}.$$

Thus $f(t) \to 1$ exponentially fast as $t \to \infty$.

Combining Steps 1–3, we conclude that $\lim_{t\to\infty} x_2(t) = x_1(0)$, which completes the proof. $\qquad\square$

We are ready to provide the proof of Proposition B.9.

***Proof of Proposition B.9.*** We proceed by induction on $L$. The case $L = 2$ has already been established in Proposition B.15. Assume now that $L \geq 3$ and that the proposition holds for $L - 1$, i.e.,

$$\sum_{i=1}^{L-1} |x_i(t) - x_1| \lesssim e^{-O(1)t},$$

and moreover $\langle x_i(t), S_{BC}x_j(t)\rangle > 0$ for all $i, j = 1, \ldots, L - 1$. We will show that $x_L(t) \to x_1$ with exponential decay rate. The proof of Proposition B.9 is divided into several claims below.

**Claim 1.** *For any $i, j$, we have $U_{ij}(t) = \langle x_i(t), S_{BC}x_j(t)\rangle > 0$ for all $t > 0$.*

*Proof of Claim 1.* Indeed, by the induction hypothesis in Proposition B.9, it suffices to consider $U_{Lj}$ with $1 \leq j \leq L$. We define the set

$$\mathcal{S} = [0, \infty)^L \subset \mathbb{R}^L.$$

By the assumption of Proposition B.9, we have

$$(U_{L1}(0), U_{L2}(0), \ldots, U_{LL}(0)) \in \mathcal{S}.$$

We now show that the trajectory

$$(U_{L1}(t), U_{L2}(t), \ldots, U_{LL}(t))$$

remains inside $\mathcal{S}$ for all $t > 0$. Suppose, for contradiction, that the trajectory leaves $\mathcal{S}$. Since it is bounded within a box, it must cross the boundary of $\mathcal{S}$ at some finite time. There are two possible scenarios:

- There exists a smallest exit time $T > 0$ and a unique index $i_\star \in [1, L]$ such that

$$U_{Li_\star}(T) = 0, \qquad U_{Li_\star}(t) > 0 \quad \text{for all } t \in (0, T),$$

  while

$$\sum_{k \neq i_\star} U_{Lk}^2(T) > 0.$$

  That is, the trajectory exits $\mathcal{S}$ through the $i_\star^{\text{th}}$ component $U_{Li_\star}$, rather than exiting all components simultaneously at time $T$.

- There exists a smallest exit time $T > 0$ such that $U_{Li}(T) = 0$ for all $i$, with $U_{Li}(t) > 0$ for all $t \in (0, T)$. In this case, the trajectory exits $\mathcal{S}$ all at once at time $T$.

We now consider the first case, when the trajectory exits $\mathcal{S}$ only at the component $i_\star$. By Lemma B.10, for $i_\star > 1$ we have

$$
\begin{aligned}
U'_{Li_\star}(T) &= \sum_{k \leq L-1} a_{Lk} U_{Lk}(T) + \sum_{k \leq i_\star - 1} a_{jk} U_{jk}(T) \left[ U_{Lk}(T) - X_{jk}(T) U_{Li_\star}(T) \right] \\
&\geq \sum_{k \leq L-1} a_{Lk} U_{Lk}(T) \\
&> 0,
\end{aligned}
$$

since at least one of the $U_{Lk}(T)$ remains strictly positive.

For $i_\star = 1$, we obtain

$$
\begin{aligned}
U'_{L1}(T) &= \sum_{j \leq L-1} a_{Lj} U_{Lj}(T) \left[ U_{j1}(T) - X_{Lj}(T) U_{L1}(T) \right] \\
&= \sum_{j \leq L-1} a_{Lj} U_{Lj}(T) U_{j1}(T) \\
&> 0.
\end{aligned}
$$

Thus, in both cases we conclude that $U'_{Li_\star}(T) > 0$.

By continuity of $U'_{Li_\star}$, there exists $\epsilon_0 > 0$ such that

$$U'_{Li_\star}(t) > 0 \quad \text{for all } t \in (T - \epsilon_0, T + \epsilon_0).$$

Hence $U_{Li_\star}(t)$ is strictly increasing on $(T - \epsilon_0, T + \epsilon_0)$, which implies

$$U_{Li_\star}(t) < U_{Li_\star}(T) = 0 \quad \forall t \in (T - \epsilon_0, T).$$

Since $U_{Li_\star}(0) > 0$, there must exist some $T_1 \in (0, T - \epsilon_0)$ such that $U_{Li_\star}(T_1) = 0$. Moreover, $U_{Li}(t) > 0$ for all $t \in [0, T_1] \subset [0, T]$. This contradicts the minimality of $T$. Therefore, the first case cannot occur.

Now we consider the second case, when the trajectory

$$(U_{L1}(t), U_{L2}(t), \ldots, U_{LL}(t))$$

exits $\mathcal{S}$ simultaneously at some time $T$. In this situation, Lemma B.10 implies that

$$U'_{Lk}(T) = 0 \qquad \text{for all } k \in [1, L].$$

Thus the trajectory passes through the equilibrium point $0 \in \mathbb{R}^L$ at time $t = T$. Choose $\epsilon_0 > 0$ sufficiently small, and consider the flow ODE for $(U_{L1}(t), U_{L2}(t), \ldots, U_{LL}(t))$ on the interval $I = (T - \epsilon_0, T + \epsilon_0)$. On this interval we observe two distinct flows satisfying the same system with the same initial data at $t = T$: the trivial solution $(U_{L1}(t), \ldots, U_{LL}(t)) \equiv 0$ and the trajectory $(U_{L1}(t), \ldots, U_{LL}(t))$ with $U_{Li}(t) > 0$ for all $t \in (T - \epsilon_0, T)$. This contradicts the uniqueness of solutions to the differential equations on $I$.

Hence the trajectory cannot exit the boundary of $\mathcal{S}$ simultaneously at all components. This completes the proof for the second case and concludes the proof of Claim 1.

$\square$

**Claim 2.** *There exist a time $T > 0$ and a constant $c_T > 0$ such that $U_{ij}(t) \geq c_T$ for all $t \geq T$ and all $1 \leq i, j \leq L$.*

*Proof of Claim 2.* For $i, j \leq L - 1$, we observe that

$$\lim_{t \to \infty} U_{ij}(t) = \lim_{t \to \infty} \langle x_i(t), S_{BC} x_j(t) \rangle = \langle x_1, S_{BC} x_1 \rangle > 0.$$

Therefore, the claim holds for such $i$ and $j$. It remains to consider the case of $U_{Lj}(t)$ for $1 \leq j \leq L$. Now, for any $i, j \leq L - 1$, there exists a time $T > 0$ such that

$$U_{ij}(t) \geq 0.9 \, (S_{BC} x_1 \cdot x_1), \quad \forall t \geq T.$$

We aim to show that for $2 \leq j \leq L$,

$$U_{Lj}(t) \geq \min \Big\{ 0.8 \, (S_{BC} x_1 \cdot x_1), \, U_{L1}(T), \ldots, U_{Lj}(T) \Big\}, \quad \forall t \geq T.$$

This will be established by induction on $j$.

For the base case $j = 1$, using Lemma B.10, we have

$$U'_{L1}(t) = \sum_{k \leq L-1} a_{Lk} U_{Lk} \big[ U_{k1} - X_{Lk} U_{L1} \big].$$

Choose any $\delta > 0$ satisfying

$$\delta < \min \big\{ 0.8 \, (S_{BC} x_1 \cdot x_1), \, U_{L1}(T) \big\}.$$

We claim that

$$U_{L1}(t) \geq \delta, \quad \forall t \geq T. \tag{30}$$

Indeed, by Lemma B.14, it suffices to check that whenever $U_{L1}(T_\star) = \delta$, we have

$$U'_{L1}(T_\star) = \sum_{k \leq L-1} a_{Lk} U_{Lk}(T_\star) \big[ U_{k1}(T_\star) - X_{Lk}(T_\star) \, \delta \big]$$

$$\geq \sum_{k \leq L-1} a_{Lk} U_{Lk}(T_\star) \big[ 0.9 \, (S_{BC} x_1 \cdot x_1) - \delta \big] > 0.$$

This establishes (30) for $j = 1$. Now, assume that for all $k \leq j - 1$,

$$\inf_{t \geq T} U_{Lk}(t) \geq \min \{ 0.8 \, \langle S_{BC} x_1, x_1 \rangle, \, U_{L1}(T), \ldots, U_{Lk}(T) \}.$$

We will show that

$$\inf_{t \geq T} U_{Lj}(t) \geq \min \{ 0.8 \, \langle S_{BC} x_1, x_1 \rangle, \, U_{L1}(T), \ldots, U_{Lj}(T) \}.$$

To this end, fix any $\delta$ satisfying

$$0 < \delta < \min \{ 0.8 \, \langle S_{BC} x_1, x_1 \rangle, \, U_{L1}(T), \ldots, U_{Lj}(T) \}.$$

We will prove that $U_{Lj}(t) \geq \delta$ for all $t \geq T$.

Recall from Lemma B.10 that

$$U'_{Lj} = \sum_{k \leq L-1} a_{Lk} U_{Lk} \left[ U_{kj} - X_{Lk} U_{Lj} \right] + \sum_{k \leq j-1} a_{jk} U_{jk} \left[ U_{Lk} - X_{jk} U_{Lj} \right].$$

Suppose $U_{Lj}(T_\star) = \delta$ for some $T_\star \geq T$. Then, we have

$$U'_{Lj}(T_\star) \geq \sum_{k \leq L-1} a_{Lk} U_{Lk}(T_\star) \left[ U_{kj}(T_\star) - \delta \right] + \sum_{k \leq j-1} a_{jk} U_{jk}(T_\star) \left[ U_{Lk}(T_\star) - \delta \right]$$

$$\geq \sum_{k \leq j-1} a_{jk} U_{jk} \left[ U_{Lk}(T_\star) - \delta \right] > 0.$$

The claim then follows by an application of Lemma B.14. $\qquad \square$

**Claim 3.** *There exist a time $T > 0$ and a constant $c_T > 0$ such that*

$$1 + X_{L1}(t) \geq c_T \quad \text{for all } t \geq T. \tag{31}$$

*Proof of Claim 3.* From Lemma B.10, we have

$$X'_{L1}(t) = a_{L1}U_{L1}(1 - X_{L1}^2)$$
$$+ \sum_{j=2}^{L-1} a_{Lj}U_{Lj}\left[X_{j1} - X_{Lj}X_{L1}\right].$$

Fix $\varepsilon \in (0, 1)$ sufficiently small. Since $X_{j1}(t) \to 1$ as $t \to \infty$ for all $j \leq L - 1$, there exists $T > 0$ such that

$$X_{j1}(t) > 1 - \varepsilon \quad \text{for all } t \geq T.$$

If $X_{L1}(t) \to 1$ as $t \to \infty$, then the claim holds trivially. Suppose, to the contrary, that $X_{L1}(t) \nrightarrow 1$ as $t \to \infty$. Then there exists an increasing sequence $\{t_n\}_{n=1}^{\infty} \subset [T, \infty)$ such that

$$X_{L1}(t_n) \leq 1 - \varepsilon.$$

We now show that not all $t_n$ can satisfy $X_{L1}(t_n) = -1$. Assume for contradiction that

$$X_{L1}(t_n) = \langle x_L(t_n), x_1 \rangle = -1 \quad \text{for all } n \geq 1.$$

This implies $x_L(t_n) = -x_1$ for all $n \geq 1$. Consequently,

$$U_{L1}(t_n) = \langle x_L(t_n), S_{BC}x_1 \rangle = -\langle x_1, S_{BC}x_1 \rangle < 0,$$

which contradicts the fact that $U_{L1}(t) > 0$ for all $t > 0$.

Therefore, there exists a time $t_\star \geq T$ such that

$$X_{L1}(t_\star) \leq 1 - \varepsilon, \quad \text{and} \quad 1 + X_{L1}(t_\star) > 0.$$

We now prove that

$$X_{L1}(t) \geq \min\{X_{L1}(t_\star),\, 1 - \varepsilon\} \quad \text{for all } t \geq t_\star,$$

which will imply inequality (31).

To this end, fix any $\delta \in (-1, \min\{X_{L1}(t_\star),\, 1 - \varepsilon\})$. We will show that $X_{L1}(t) \geq \delta$ for all $t \geq t_\star$ by applying Lemma B.14.

Indeed, suppose $X_{L1}(t) = \delta$ for some $t \geq t_\star$. Then, using the earlier lower bounds and the fact that $U_{ij}(t) > 0$, we obtain

$$X'_{L1}(t) \geq a_{L1}U_{L1}(1 - \delta^2) + \sum_{j=2}^{L-1} a_{Lj}U_{Lj}\left[(1 - \varepsilon) - \delta\right] > 0.$$

This verifies the condition of Lemma B.14, completing the proof of Claim 3. $\qquad\square$

**Claim 4.** *There exist a time $T > 0$ and a constant $c_T > 0$ such that for all $t \geq T$,*

$$1 - X_{L1}(t) \lesssim e^{-c_T t}.$$

*Proof of Claim 4.* By the preceding claims and the induction hypothesis, there exist $T > 0$ and $c_T > 0$ such that for all $t \geq T$,

$$1 + X_{L1}(t) \geq c_T, \quad U_{ij}(t) \geq c_T, \quad \max_{2 \leq j \leq L-1}(1 - X_{j1}(t)) \lesssim e^{-c_T t}.$$

For $t \geq T$, we compute

$$\frac{d}{dt}\left[1 - X_{L1}(t)\right] = -X'_{L1}(t) = -a_{L1}U_{L1}(1 - X_{L1}^2) - \sum_{j=2}^{L-1} a_{Lj}U_{Lj}\left[X_{j1} - X_{Lj}X_{L1}\right]$$

$$= -a_{L1}U_{L1}(1 - X_{L1}^2) - \sum_{j=2}^{L-1} a_{Lj}U_{Lj}(X_{j1} - 1) - \sum_{j=2}^{L-1} a_{Lj}U_{Lj}(1 - X_{Lj}X_{L1})$$

$$= -a_{L1}U_{L1}(1 - X_{L1})(1 + X_{L1}) + \sum_{j=2}^{L-1} a_{Lj}U_{Lj}(1 - X_{j1}) - \sum_{j=2}^{L-1} a_{Lj}U_{Lj}(1 - X_{Lj}X_{L1}).$$

Note that the third term is non-positive, since $|X_{Lj}X_{L1}| \leq 1$, so we may drop it to obtain an upper bound:

$$\frac{d}{dt}\left[1 - X_{L1}(t)\right] \leq -a_{L1}U_{L1}(1 + X_{L1})(1 - X_{L1}) + \sum_{j=2}^{L-1} a_{Lj}U_{Lj}(1 - X_{j1})$$

$$\leq -c_1(1 - X_{L1}(t)) + C_0 e^{-c_T t},$$

for some positive constants $c_1, C_0 > 0$, where we have used the lower bounds $U_{L1}(t) \geq c_T$, $1 + X_{L1}(t) \geq c_T$, and the exponential decay $1 - X_{j1}(t) \lesssim e^{-c_T t}$.

Applying the Gronwall inequality to this differential inequality yields

$$1 - X_{L1}(t) \lesssim e^{-\min\{c_1, c_T\}t},$$

which completes the proof of Claim 4. $\qquad\square$

Proposition B.9 now follows from Claims 1–4. $\qquad\square$

We end this subsection by the proof of Theorem B.8.

***Proof of Theorem B.8.*** By Proposition B.15 and Proposition B.11, the second token converges to $\pm x_1(0)$. Moreover, by Proposition B.11, there exists a minimal integer $k \in [3, L]$ such that, after possibly replacing each $x_i$ by its reflection $\widetilde{x}_i = -x_i$ for $i \leq k - 1$, one of the following two cases holds:

(a) $\sum_{i=2}^{k-1} |\widetilde{x}_i(t) - x_1(0)| + \left|\langle x_k(t), S_{BC}x_1(0)\rangle\right| \lesssim e^{-O(1)t},$

(b) $\sum_{i=2}^{k-1} |\widetilde{x}_i(t) - x_1(0)| + |\widetilde{x}_k(t) - x_1(0)| \lesssim e^{-O(1)t}.$

It therefore suffices to consider the case $k < L$ and case (a). In this scenario, we have

$$\left|\langle x_k(t), S_{BC}x_1(0)\rangle\right| \lesssim e^{-O(1)t}.$$

This implies that $x_k(t)$ converges (along a subsequence) to some unit vector $\xi \in \langle S_{BC}x_1(0)\rangle^{\perp}$. By Proposition B.13, the set of initial configurations for which such a limit $\xi$ occurs is of measure zero in the phase space $(\mathbb{S}^{D-1})^{k-1}$. In particular, $x_k \to \xi \notin \{\pm x_1(0)\}$, and the subsequent tokens $x_{k+1}, \ldots, x_L$ evolve under the influence of $x_k$ converging to $\xi$. It follows that the entire configuration $(x_1, \ldots, x_L)$ converges to a point outside $\Omega_0$, and such initial conditions form a set of measure zero in $(\mathbb{S}^{D-1})^L$. This completes the proof. $\qquad\square$

### B.3 BEYOND MAMBA: AUTOREGRESSIVE SEQUENCE MODELS

**Theorem B.16.** *Let $\{x_i\}_{i=1}^{L} \subset \mathbb{S}^{D-1}$ solve the system*

$$x_i' = \sum_{j=1}^{i} \mathbf{P}_{x_i}\left(a_{ij}x_j\right), \quad 1 \leq i \leq L,$$

*where the coefficients $a_{ij} \in \mathbb{R}_+$ are positive continuous functions depending on $\vec{x} = (x_1, x_2, \cdots, x_L) \in (\mathbb{S}^{D-1})^L$. Assume that $\langle x_i(0), x_1(0) \rangle \notin \{\pm 1\}$ for all $i \geq 2$. Then, for almost all initial data, there exists a constant $C_0 > 0$ such that for all $t > 0$,*

$$\sum_{i=1}^{L} |x_i(t) - x_1(0)| \lesssim e^{-C_0 t}.$$

**Remark B.17.** This result can be viewed as the analogue of Theorem B.8 in the setting of positive couplings. The positivity of the coefficients $a_{ij}$ enforces a natural alignment mechanism among the tokens and greatly simplifies the analysis. In particular, we obtain a uniform exponential decay rate toward the consensus configuration. Once the system is close to consensus, the nonlinear interactions reduce to their linearization, so the long-time dynamics are effectively linear. Furthermore, any alternative limit points capable of attracting trajectories must also do so at an exponential rate, but such cases can occur only on a negligible set of initial conditions. This mirrors the behavior already observed in the Mamba token model, though here it arises in a more transparent form.

*Proof.* We proceed by induction on $L$.

For the first token, we have

$$x_1' = \mathbf{P}_{x_1}(a_{11}x_1) = a_{11}\mathbf{P}_{x_1}(x_1) = 0,$$

so $x_1(t) = x_1(0)$ for all $t > 0$.

For the second token,
$$x_2' = a_{21}\mathbf{P}_{x_2}(x_1).$$
We show that $x_2(t) \to x_1(0)$ as $t \to \infty$. Define $f(t) = \langle x_2(t), x_1(0) \rangle$. Then

$$f'(t) = \partial_t x_2 \cdot x_1 = a_{21}\left[x_1 - (x_1 \cdot x_2)x_2\right] \cdot x_1 = a_{21}\left[1 - f(t)^2\right].$$

Since $f(0) \in (-1, 1)$, we have $f(t) \in (-1, 1)$ for all $t \geq 0$, and hence $f'(t) > 0$. Thus $f$ is strictly increasing and converges to some $\theta \in (-1, 1]$. For large $t$,

$$\frac{d}{dt}(1 - f(t)) = -f'(t) = -a_{21}(1-f)(1+f) \lesssim -(1+\theta)(1-f).$$

By Gronwall's inequality, $1 - f(t) \lesssim e^{-(1+\theta)t}$, which proves the case $L = 2$.

Now assume by induction that for some $i \geq 3$,

$$\sum_{k=1}^{i-1} |x_k(t) - x_1(0)| \lesssim e^{-O(1)t}.$$

We show that $|x_i(t) - x_1(0)| \lesssim e^{-O(1)t}$. Let $f_i(t) = \langle x_i(t), x_1(0) \rangle$ and $g_{ij}(t) = \langle x_i(t), x_j(t) \rangle$. Then

$$f_i'(t) = \partial_t x_i \cdot x_1 = \sum_{j \leq i-1} a_{ij}\left[x_j - (x_i \cdot x_j)x_i\right] \cdot x_1$$

$$= \sum_{j \leq i-1} a_{ij}\left[f_j(t) - g_{ij}(t)f_i(t)\right]$$

$$= a_{i1}(1 - f_i^2) + \sum_{2 \leq j \leq i-1} a_{ij}\left[f_j(t) - g_{ij}(t)f_i(t)\right].$$

We consider two cases:

- **Case 1:** There exists a strictly increasing sequence of times in $\{t \geq 0 : f_i(t) + 1 > 0\}$.

- **Case 2:** $f_i(t) \to -1$ as $t \to \infty$.

**Case 1:** By the induction hypothesis, $f_j(t) \to 1$ for $j \leq i - 1$. Hence, there exists $T > 0$ such that $f_j(t) \geq 0.9$ for all $t \geq T$ and $j \leq i - 1$. Let $t_\star \geq T$ be such that $f_i(t_\star) + 1 > 0$. We show that

$$f_i(t) \geq \min\{0.8, f_i(t_\star)\} \quad \forall t \geq t_\star. \tag{32}$$

To prove this, let $-1 < \delta < \min\{0.8, f_i(t_\star)\}$. Using Lemma B.14, we verify that if $f_i(t) = \delta$, then

$$f_i'(t) \geq a_{i1}(1 - \delta^2) + \sum_{2 \leq j \leq i-1} a_{ij}(0.9 - \delta) > 0,$$

which implies (32). Now, for large $t$,

$$\frac{d}{dt}(1 - f_i) = -f_i' = -a_{i1}(1 - f_i^2) - \sum_{2 \leq j \leq i-1} a_{ij} \left[ f_j - g_{ij} f_i \right]$$

$$\leq -a_{i1}(1 - f_i)(1 + f_i) + \sum_{j \leq i-1} a_{ij}(1 - f_j)$$

$$\lesssim -(1 + f_i(t_\star))(1 - f_i) + e^{-O(1)t}.$$

Gronwall's inequality then gives $1 - f_i(t) \lesssim e^{-O(1)t}$.

**Case 2:** Suppose $f_i(t) \to -1$. Then $x_i(t) \to -x_1(0)$, and for $j \leq i - 1$ we have $g_{ij}(t) \to -1$ exponentially. From the equation

$$f_i'(t) = a_{i1}(1 - f_i)(1 + f_i) + \sum_{2 \leq j \leq i-1} a_{ij} \left[ f_j(t) - g_{ij}(t) f_i(t) \right],$$

we derive, for large $t$,

$$\frac{d}{dt}(1 + f_i(t)) \geq c_0(1 + f_i) - C_\sigma e^{-\sigma t},$$

for some $c_0, C_\sigma, \sigma > 0$. Consequently, integrating this inequality and using the boundedness of $1 + f_i(t)$ yields

$$1 + f_i(t) \lesssim e^{-O(1)t},$$

so $|x_i(t) + x_1(0)| \lesssim e^{-O(1)t}$.

Finally, we show that the set of initial data for which $x_i \to x_1(0)$ for $i \leq L - 1$ and $x_L \to -x_1(0)$ has measure zero. Define perturbations:

$$\begin{cases} x_i = x_1 + \delta \widetilde{x}_i, & 2 \leq i \leq L - 1, \\ x_L = -x_1 + \delta \widetilde{x}_L, \\ \widetilde{f}_i(t) = \langle \widetilde{x}_i(t), x_1 \rangle. \end{cases}$$

From the equation for $f_L'$, we obtain the linearized dynamics:

$$\partial_t \widetilde{f}_L = 2 \left[ \sum_{j=1}^{L-1} a_{Lj}(x_\star) \right] \widetilde{f}_L,$$

where $x_\star = (x_1, \ldots, x_1, -x_1)$. The coefficient

$$\lambda_+ = 2 \sum_{j=1}^{L-1} a_{Lj}(x_\star) > 0$$

is a positive eigenvalue, hence the equilibrium is linearly unstable. By decomposing

$$\begin{cases} \widetilde{x}_L = \widetilde{f}_L(t) x_1 + \widetilde{x}_L^\perp, & \widetilde{x}_L^\perp \perp x_1, \\ \widetilde{x}_j = \widetilde{f}_j(t) x_1 + \widetilde{x}_j^\perp, & \widetilde{x}_j^\perp \perp x_1, \quad j \leq L - 1, \end{cases}$$

we observe that the full Jacobian at $x_\star$ also possesses the positive eigenvalue $\lambda_+$, producing linear instability in the $x_1$ direction. As in the proof of Proposition B.13, the corresponding initial conditions form a set of measure zero. This completes the proof.

$\square$

# C  EXPERIMENT - ADDITIONAL DETAILS

## C.1  PREDICT ATTENTION SINK

**Experimental Setup.**  We adopt the threshold-based metric of Xiao et al. (2024), adapted to the hidden attention structure of SSMs. For a given layer $\ell \in \{1, \dots, L\}$ and channel $c$, let $A^{\ell,c} \in \mathbb{R}^{T \times T}$ denote the hidden attention matrix, where $T$ is the sequence length. The importance of token $k$ in $(\ell, c)$ is defined as

$$\alpha_k^{\ell,c} = \frac{1}{T - k + 1} \sum_{i=k}^{T} A_{i,k}^{\ell,c}. \tag{33}$$

We say that channel $(\ell, c)$ exhibits an attention sink at the first token if

$$\alpha_1^{\ell,c} > \varepsilon, \tag{34}$$

where $\varepsilon$ is a fixed threshold (we set $\varepsilon = 0.3$ in our experiments).

The *layer-wise sink rate* is given by

$$\text{Sink}_\ell = \frac{1}{H_s} \sum_{c \in \mathcal{C}_\ell} \mathbf{1}\left(\alpha_1^{\ell,c} > \varepsilon\right), \tag{35}$$

where $\mathcal{C}_\ell$ is a subset of channels sampled from layer $\ell$, and $H_s = |\mathcal{C}_\ell|$. Finally, the *model-level sink score* is

$$\text{Sink} = \frac{1}{L} \sum_{\ell=1}^{L} \text{Sink}_\ell. \tag{36}$$

**Evaluation.**  To empirically estimate these quantities, we evaluate the Falcon-Mamba-7B model (Zuo et al., 2024) on the dataset of Xiao et al. (2024), which consists of 170 samples of length $T = 16$. Since each layer contains 4096 channels, we uniformly sample $H_s = 200$ channels per layer to estimate $\text{Sink}\ell$ and the overall Sink. We then report $\text{Sink}\ell$ as a function of layer depth and the aggregate Sink across tokens in Figure 3. For visualization, we additionally average the softmax-normalized attention matrices $\text{softmax}(A^{\ell,c})$ across the samples to obtain per-layer attention maps, as shown in Figure 2.

## C.2  RANDOMLY REORDERING TOKENS TO DIVERSIFY CLUSTERING EFFECT

**Dataset.**  We evaluate our random reordering strategy on the ImageNet-1K dataset (Deng et al., 2009), which consists of 1.28M training images and 50K validation images across 1,000 categories.

**Hyperparameters.**  For training, we adopt the same configuration as in our baseline MambaVision-T model (Hatamizadeh & Kautz, 2024). We use AdamW as the optimizer with peak learning rate $8 \times 10^{-4}$, momentum parameters $\beta_1 = 0.9, \beta_2 = 0.999$, and weight decay $0.05$. Training is conducted with a batch size of 512 and linear warmup followed by cosine decay for the learning rate schedule. Dropout is set to 0.0. The full hyperparameter configuration is listed in Table 2 for reproducibility.

Table 2: Hyperparameters configuration for Image Classification with random token reordering on ImageNet-1K.

| | |
|---|---|
| Optimizer | AdamW |
| Peak Learning Rate | 0.0008 |
| Momentum | $\beta_1 = 0.9, \ \beta_2 = 0.999$ |
| Weight Decay | 0.05 |
| Batch Size | 512 |
| Learning Rate Warmup | Linear |
| Learning Rate Scheduler | Cosine decay |
| Dropout | 0.0 |

**Training details.**    We train the tiny version (31.8M parameters) of MambaVision with and without random token reordering. The reordering is performed by sampling a random permutation of tokens before each forward pass, ensuring that the identity of the first token varies. This prevents collapse toward a fixed attractor and diversifies the clustering dynamics as predicted by our theoretical framework. We follow the same training procedure as in (Yang et al., 2021; Hatamizadeh & Kautz, 2024; Hatamizadeh et al., 2023) and ensure that the two settings (baseline and reordered) differ only in the reordering operation. Top-1 and Top-5 classification accuracy are reported in Table 1 and Figure 4.

