# OpenReview forum: "Token Dynamics on Spheres in Mamba Models"
_ICLR.cc/2026/Conference — Submitted to ICLR 2026_

### Official Review · Reviewer_ZK7P · 2025-10-23

**Soundness:** 2
**Presentation:** 3
**Contribution:** 1
**Rating:** 2
**Confidence:** 3

**Summary:**

This paper investigates the token dynamics of autoregressive sequence models (mainly on Mamba) by treating the layers’ index as a continuous variable. Following the framework of [Karagodin et al., 2024], the authors model the token dynamics of Mamba as the solution to an ordinary differential equation, where the continuous layer index is interpreted as time $t$. They analyze the limit as $t \to \infty$ and, under certain assumptions, prove that (1) the first token converges to a vector with specific properties, (2) all subsequent tokens are attracted to the first token, and (3) this convergence is exponential. Based on these results, they conclude that tokens form a small number of clusters and present two applications derived from this finding.

**Strengths:**

1. The paper provides a theoretical analysis of token dynamics based on the existing framework (Eq. 6, 7), and these findings are also verified empirically through simulations.
2. The study demonstrates exponential convergence in token dynamics, which clearly distinguishes this paper from prior research.

**Weaknesses:**

1. The theoretical assumptions are strong, and there is no empirical justification for them. The paper proceeds to theoretically justify the observations shown in Figure 1. However, the figure’s results are (1) qualitative rather than quantitative, (2) based on small-scale settings, and (3) constrained by random parameter choices.
2. The range of applications is limited. The insights into token dynamics are applied only to (1) understanding the attention sink phenomenon and (2) improving performance via token permutation during training. Regarding (2), as noted in (Weakness 4), the justification is insufficient.
3. In Section 4.1, the paper claims to provide a theoretical explanation of the attention sink (line 374), yet it never explicitly explains how the attraction of later tokens to the first token causes the attention sink phenomenon.
4. The rationale behind the method proposed in Section 4.2 is weak. The fact that tokens form a small number of clusters does not directly imply that randomly permuting token order during training will improve performance. Moreover, the experiment is small-scale, and the final accuracy hardly changes with or without the proposed technique (Figure 4). Since no error bars are reported, the generality of this result remains unclear.
5. The normalization used in the paper deviates from RMSNorm, yet the authors do not investigate how this deviation affects their conclusions. Although the paper emphasizes as a contribution that it includes the effect of layer normalization (lines 58 and 73), the analysis presented here is insufficient to support that claim.
6. Assuming that parameters do not depend on $t$ ignores the differences between layers. It is unclear what practical implications or benefits such an observation offers for real-world implementations.

**Questions:**

1. The paper extends the theoretical results developed for Mamba to a broader class of autoregressive sequence models. What are the statements that can be proved for Mamba but not for this broader class?
2. Can the results of this study be used to explain differences in token dynamics across models?
3. Why does Figure 1 not include the results for the case where the parameter $a$ depends on time?
4. As noted in Remark 3.4, the result of Theorem 3.2 is consistent with experiments, while Theorem 3.3 is not. Why does this inconsistency arise only in the two-dimensional case?
5. Theorem 3.5 and Proposition 3.6 state that, as $t \to \infty$, all tokens converge to the first token. However, Figure 1 and Remark 3.7 demonstrate that “tokens in a deep Mamba model tend to form a small number of clusters.” Could the authors explain this apparent discrepancy?

---

> ### Author Response · Authors · 2025-11-21
> **Reply [1/3]**
>
> Thank you for your thoughtful review and valuable feedback. Below we address your concerns.
>
> **W1. The theoretical assumptions are strong, and there is no empirical justification for them. The paper proceeds to theoretically justify the observations shown in Figure 1. However, the figure’s results are (1) qualitative rather than quantitative, (2) based on small-scale settings, and (3) constrained by random parameter choices.**
>
> **W6. Assuming that parameters do not depend on $t$ ignores the differences between layers. It is unclear what practical implications or benefits such an observation offers for real-world implementations.**
>
> **Answer W1 and W6.** While your concern about the simplified setting used in our theory is valid, it is standard in theoretical work to adopt simplified architectures in order to make rigorous analysis feasible, while still retaining enough structure to capture the key behaviors observed in practice. Several high-quality related works by leading researchers follow this approach, for example:
>
> [B] Karagodin et al., Clustering in Causal Attention Masking, NeurIPS 2024.
>
> [C] Geshkovski et al., The Emergence of Clusters in Self-Attention Dynamics, NeurIPS 2024.
>
> [D] Geshkovski et al., A Mathematical Perspective on Transformers, Bulletin of the AMS 2024.
>
> [E] Vo et al., Demystifying the Token Dynamics of Deep Selective State Space Models, ICLR 2025 (spotlight).
>
> All above papers using time-independent parameters and one key components (either attention in Transformer or S6 in Mamba).
>
> Our results generalize the findings in [E] by incorporating layer normalization and studying token dynamics on high-dimensional spheres. We further extend these results to general autoregressive sequence models, which we view as a substantial and nontrivial generalization.
>
> Handling models with time-dependent parameters and additional practical components (such as multiple normalization layers, convolutions, or gating mechanisms) remains extremely challenging, and to the best of our knowledge, no existing theoretical work has achieved this. We agree that this is an important direction and leave it for future research.
>
>
> **W2. The range of applications is limited. The insights into token dynamics are applied only to (1) understanding the attention sink phenomenon and (2) improving performance via token permutation during training. Regarding (2), as noted in (Weakness 4), the justification is insufficient.**
>
> **Answer W2.** Our main goal is theoretical, not to exhaust all possible applications. The two applications we show:
>
> (1) explaining the attention sink and,
> (2) a token-permutation trick
>
> are meant as illustrative examples of how the theory can be used.
>
> For (1), we provide the first rigorous mathematical explanation of the sink phenomenon in SSMs, which we view as a central contribution in itself. For (2), the permutation experiment is intentionally simple: it serves as a minimal empirical check that the first-token attractor predicted by our theory has a measurable effect on training. The concern about its justification is addressed in our reply to Weakness 4; here we emphasize that this experiment is supportive, not the core contribution of the paper.
>
>
> **W3. In Section 4.1, the paper claims to provide a theoretical explanation of the attention sink (line 374), yet it never explicitly explains how the attraction of later tokens to the first token causes the attention sink phenomenon.**
>
> **Answer W3.** Theorem 3.3 shows that later tokens are pulled toward the limit point of the first token. As a consequence, attention weights concentrate on those of the first token over depth. Thus, the attraction phenomenon proved in our theoretical analysis is precisely the mechanism that produces the attention sink in practice. We have added this explanation to the revised version of the paper.

---

> ### Author Response · Authors · 2025-11-21
> **Reply [2/3]**
>
> **W4. The rationale behind the method proposed in Section 4.2 is weak. The fact that tokens form a small number of clusters does not directly imply that randomly permuting token order during training will improve performance. Moreover, the experiment is small-scale, and the final accuracy hardly changes with or without the proposed technique (Figure 4). Since no error bars are reported, the generality of this result remains unclear.**
>
> **Answer Q4.** Our main goal in this paper is theoretical: to analyze token dynamics and rigorously explain the first-token attractor and clustering phenomenon. The experiment in Section 4.2 is only a side application, whose purpose is to illustrate that the theoretically predicted clustering of tokens into a small number of attractors has concrete consequences and that a very simple, theory-motivated intervention (random permutation of tokens) can partially mitigate it.
>
> In our setup, the number of layers is large enough, so the discrete-depth dynamics are well approximated by the continuous-time limit studied in our theory. Under this regime, random permutation of token order during training does break the persistent identity of the first token and therefore disrupts the clustering pattern predicted by the continuous-time analysis. This is exactly what we see empirically: although the performance gain is modest, the clustering is weakened and accuracy improves slightly yet consistently.
>
> We will make this limited, illustrative role of Section 4.2 clearer in the revision and avoid overselling the empirical impact.
>
> **W5. The normalization used in the paper deviates from RMSNorm, yet the authors do not investigate how this deviation affects their conclusions. Although the paper emphasizes as a contribution that it includes the effect of layer normalization (lines 58 and 73), the analysis presented here is insufficient to support that claim.**
>
> **Anwer W5.** We clarify that in our theoretical analysis we do use RMSNorm (in its continuous-time limit form), not an arbitrary ad-hoc normalization.
>
> Our approach follows previous theoretical works [B,C,D] on token dynamics: by modeling RMSNorm, one can represent the normalization step in the continuous-time limit as a projection operator onto a sphere. This spherical projection is exactly what we exploit in our proofs to control the norms and geometry of the token trajectories.
>
> In practice, we observe the same qualitative behavior (first-token attraction and clustering) both with RMSNorm and with LayerNorm. Our conclusions rely on the induced near-spherical constraint on token norms, which is shared by both normalizations, rather than on fine implementation details of a specific variant. We will clarify this point in the revised version and adjust the wording around lines 58 and 73 accordingly.
>
>
>
> **Q1. The paper extends the theoretical results developed for Mamba to a broader class of autoregressive sequence models. What are the statements that can be proved for Mamba but not for this broader class?**
>
> **Answer Q1.** For a broader class of autoregressive sequence models, we prove a sufficient condition for the one-cluster phenomenon (Theorem 3.9). We have not yet proved the more general case in which the number of clusters can be greater than one for these broader classes (Theorems 3.2, 3.3, and 3.5).
>
>
> **Q2. Can the results of this study be used to explain differences in token dynamics across models?**
>
> **Answer Q2.** No, the results of this study cannot be used to explain differences in token dynamics across models. The issue is that different models with different settings lead to very different dynamical systems, which may require quite different techniques and levels of difficulty to analyze. Within the scope of our paper, we provide a rigorous theoretical analysis for the Mamba model. In addition, we provide a proof of the one-clustering phenomenon for a broader class of autoregressive sequence models.
>
> **Q3. Why does Figure 1 not include the results for the case where the parameter $a$ depends on time?**
>
> **Answer Q3.** We thank the reviewer for pointing this out. We have conducted an additional simulation where the parameter $a$ depends on time, and the results observed were consistent with those presented in the original analysis, that is, tokens converge to points at some coordinate unit vectors $\pm e_d$. This update has been reflected in the revised version of the paper.

---

> ### Author Response · Authors · 2025-11-21
> **Reply [3/3]**
>
> **Q4. As noted in Remark 3.4, the result of Theorem 3.2 is consistent with experiments, while Theorem 3.3 is not. Why does this inconsistency arise only in the two-dimensional case?**
>
> **Answer Q4.** We thank the referee for raising this question and for allowing us to clarify a potential source of confusion. Both Theorem 3.2 and Theorem 3.3 concern the **first token**. The difference between them arises from the assumptions placed on $S_{\Delta}$ and from the special geometry available only in two dimensions.
>
> Theorem 3.2 applies in **all dimensions** $D \ge 2$. Under the structural assumption
> $$
> S_{\Delta,m} = \mathrm{sign}(y_m(0))\, e_m,
> $$
> the dynamics of the first token simplify considerably: the coordinate with the largest initial magnitude dominates the evolution. Consequently, in every dimension the first token converges to the corresponding coordinate direction. For instance, in $D=3$, the limit must be one of
> $$(\pm1,0,0),\ (0,\pm1,0),\ (0,0,\pm1).$$
> Thus, Theorem 3.2 establishes a universal clustering phenomenon for the first token valid in any dimension once the $S_\Delta$ matrices align with the initial signs.
>
> Theorem 3.3, on the other hand, is a result **specific to the two-dimensional case** and again concerns the first token, but **without** imposing any structural assumptions on $S_\Delta$. When $D=2$, we can express $y(t)$ in polar coordinates and reduce the first-token dynamics to a scalar ODE for the angle. This explicit reduction allows us to classify **all** possible limit points. In this unrestricted 2D setting, special geometric cancellations may occur, producing additional stationary points on the circle that do not arise in higher dimensions.
>
> In summary:
> — Both theorems describe the asymptotic behavior of the **first token**.
> — Theorem 3.2 provides a dimension-independent convergence result under an alignment assumption on $S_\Delta$.
> — Theorem 3.3 reveals a richer set of possible limit points only in 2D when that assumption is removed and the dynamics reduce to an explicit angular ODE.
>
> We hope this explanation resolves the apparent discrepancy.
>
>
> **Q5. Theorem 3.5 and Proposition 3.6 state that, as $t\to\infty$, all tokens converge to the first token. However, Figure 1 and Remark 3.7 demonstrate that “tokens in a deep Mamba model tend to form a small number of clusters.” Could the authors explain this apparent discrepancy?**
>
> **Answer Q5.** There is no contradiction between Theorem 3.5/Proposition 3.6 and the clustering behavior shown in Figure 1 and Remark 3.7.
>
> - Theorem 3.5 states that, under certain assumptions, each token $x_i(t)$ converges exponentially either to $x_1(0)$ or to $-x_1(0)$, depending on the sign of the corresponding coefficient $s_i$. Thus, in this setting there are *two possible clusters*: one around $x_1(0)$ and one around $-x_1(0)$.
>
> - Proposition 3.6 imposes stricter assumptions under which all tokens converge exponentially to $x_1(0)$. In this case, there is *only one cluster*.
>
> - In practice, when the assumptions of Theorem 3.5 do not fully hold, it is possible to observe more than two clusters at the end of the dynamics (see for examples in Figure 1(a)). However, this occurs only rarely. Almost all trajectories still fall into one of the two dominant clusters predicted by Theorem 3.5.
>
> We have clarified in the revised version that Theorem 3.5 predicts (up to sign) two clusters, Proposition 3.6 predicts one cluster, and the empirical figures are fully consistent with these theoretical regimes.
>
> ---
> We thank the reviewer for the constructive feedback and thoughtful suggestions. If our revisions and responses have resolved the concerns, we would respectfully request that the overall score be raised accordingly. We remain fully open to further discussion in the next stage of the review process.

---

### Official Review · Reviewer_FDLD · 2025-11-02

**Soundness:** 4
**Presentation:** 3
**Contribution:** 2
**Rating:** 2
**Confidence:** 4

**Summary:**

The paper examines Mamba models through the lens of dynamical systems. The authors primarily aim to demonstrate that the first token functions as an attractor for the subsequent tokens.

**Strengths:**

1. The paper seeks to analyze the token dynamics of Mamba models and explore the concept of sinks within them.

2. It also aims to identify practical applications or benefits that arise from the theoretical framework it develops.

**Weaknesses:**

1. The paper makes several assumptions that seem insufficiently supported (see questions for details).

2. A few figures in the paper are difficult to interpret (see questions).

3. The usefulness of Section 4.2 is unclear to me (see questions).

4. Some of the claims in the paper appear somewhat overstated (see questions).

**Questions:**

1. The abstract highlights the role of layer normalization; however, most of the proofs appear to rely on the L2 norm. It may be helpful to reference prior works such as [A] and [B], which discuss constructions that modify layer normalization to effectively reduce to an L2 norm. The authors might also consider clarifying where exactly layer normalization is applied within the SSM layer and how it interacts with the overall model dynamics.

2.  The reasoning underlying Proposition 1 could benefit from further explanation. Since the system does not necessarily reach equilibrium, it would be helpful for the authors to clarify the intuition and potential implications of the result in cases where equilibrium is not achieved.

3. It would strengthen the paper to discuss whether some of the stated assumptions could be empirically validated. For example, do these assumptions hold in trained SSMs or when SSMs are trained from scratch? Clarifying which assumptions align with observed behavior in practical settings would make the theoretical contributions more compelling and relevant to real-world machine learning models.

4. The motivation for focusing on asymptotic analysis as T -> $\infty$ could be elaborated upon. Since most networks are of finite depth (often fewer than 100 layers), it might be helpful to comment on whether finite-time behavior or convergence rates also play a meaningful role in the theoretical conclusions. (I understand that there are other works that have used similar analysis techniques but I'm trying to question the analysis technique as a whole).

5. The purpose and practical relevance of Section 4.2 could be clarified. The reported improvements appear modest and potentially within the range of standard deviation, and it is not immediately clear how the random perturbation setup extends to language modeling. Providing additional context or justification for this section would enhance its contribution to the overall narrative.

6. The explanation of Figure 1 could be made clearer. It would be useful to specify what is meant by “most trajectories converge”,  for example, whether this refers to a percentage, threshold, or specific criterion. Detailing the initialization scheme and other experimental settings used to produce the figure would also improve interpretability.

7. The paper could benefit from extending Figure 2 to additional channels and layers. It would also be valuable to describe the experimental setup more thoroughly, including dataset details, model configuration, training protocol, and hyperparameters, to facilitate reproducibility and interpretation.

8. Equations 1619-1620 may contain a typographical error. The authors are kindly requested to verify and, if necessary, correct this in the final version.

---
**Minor Typos**

Line 284: far more challenge -> far more challenging

---

**References**

[A] Rajaraman, N., Bondaschi, M., Makkuva, A. V., Ramchandran, K., & Gastpar, M. (2024). Transformers on markov data: Constant depth suffices. Advances in Neural Information Processing Systems, 37, 137521-137556.

[B] Ekbote, C., Makkuva, A. V., Bondaschi, M., Rajaraman, N., Gastpar, M., Lee, J. D., & Liang, P. P. (2025). What one cannot, two can: Two-layer transformers provably represent induction heads on any-order Markov chains. In Proceedings of the 39th Annual Conference on Neural Information Processing Systems (NeurIPS 2025).

---

> ### Author Response · Authors · 2025-11-21
> **Reply [1/3]**
>
> Thank you for your thoughtful review and valuable feedback. Below we address your concerns.
>
> **Q1. [...] It may be helpful to reference prior works such as [A] and [B], which discuss constructions that modify layer normalization to effectively reduce to an L2 norm. The authors might also consider clarifying where exactly layer normalization is applied within the SSM layer and how it interacts with the overall model dynamics.**
>
> **Answer Q1.** We follow the setting typically used in the theoretical analysis of token dynamics in previous papers such as [C, D, E]. In these settings, we use the L2 norm. We have clarified this point in the revised version and now explicitly refer to the prior works [A, B].
>
> [C] Karagodin et al., Clustering in Causal Attention Masking, NeurIPS 2024.
>
> [D] Geshkovski et al., The Emergence of Clusters in Self-Attention Dynamics, NeurIPS 2024.
>
> [E] Geshkovski et al., A Mathematical Perspective on Transformers, Bulletin of the AMS 2024.
>
> **Q2. The reasoning underlying Proposition 1 could benefit from further explanation. Since the system does not necessarily reach equilibrium, it would be helpful for the authors to clarify the intuition and potential implications of the result in cases where equilibrium is not achieved.**
>
> **Answer Q2.** There is no Proposition 1 in our paper. We assume that you are discussing about Proposition 3.1 instead. We agree that the reasoning underlying Proposition 3.1 deserves a clearer explanation.
>
> First, we explicitly state in the paper that Proposition 3.1 is a characterization of the set of equilibrium points. It is not a convergence result, and in particular not every equilibrium point is a limit point of the dynamics. The actual limit points are characterized later, in Theorems 3.2 and 3.3, where we use a different and more innovative approach to identify which equilibria can be attained in the long-time limit. Thus, Proposition 3.1 and Theorems 3.2–3.3 play complementary roles: Proposition 3.1 describes all equilibria; Theorems 3.2–3.3 describe the reachable ones.
>
> Intuitively, this has a stability interpretation:
>
> - Initial conditions starting near an equilibrium that is a limit point will remain stable (or metastable) even under perturbations/noise.
>
> - Initial conditions starting near an equilibrium that is not a limit point will exhibit unstable behavior in the presence of noise and will typically be pushed away.
>
> In this sense, Proposition 3.1 also describes the direction of drift of the dynamics: trajectories are driven toward the region of state space described there, and the attracting effect of the first token is present even in cases where the system does not converge to a single equilibrium. We will clarify this intuition and the connection to Theorems 3.2–3.3 in the revised version.
>
> **Q3. It would strengthen the paper to discuss whether some of the stated assumptions could be empirically validated [...]**
>
> **Answer Q3.** We understand the concern that our simplified setting may appear less practical. However, the scope of this paper is purely theoretical, and its goal is not to analyze the full practical architecture, but to provide a rigorous mathematical explanation for phenomena observed in practice under a simplified, yet still meaningful, setting. This is a different goal from that of an empirical paper, whose primary focus would be to validate assumptions on large-scale trained models.
>
> It is standard in theoretical work to adopt reduced architectures in order to make rigorous analysis feasible while retaining enough structure to capture key behaviors. Several high-quality theoretical works follow exactly this approach, for example:
>
> [B] Karagodin et al., Clustering in Causal Attention Masking, NeurIPS 2024.
>
> [C] Geshkovski et al., The Emergence of Clusters in Self-Attention Dynamics, NeurIPS 2024.
>
> [D] Geshkovski et al., A Mathematical Perspective on Transformers, Bulletin of the AMS 2024.
>
> [E] Vo et al., Demystifying the Token Dynamics of Deep Selective State Space Models, ICLR 2025 (spotlight).
>
> All of these works use time-independent parameters and focus on a single component (attention in Transformers or S6 in Mamba), exactly as we do.
>
> Our results generalize [E] by incorporating layer normalization and studying token dynamics on high-dimensional spheres, and we further extend the theory to general autoregressive sequence models. We view this as a substantial and nontrivial theoretical step.
>
> We agree that it is interesting and important to empirically test how well our assumptions match trained SSMs in practice (e.g., approximate norm preservation and bounded effective couplings), but a thorough empirical validation belongs to a different type of paper with a different scope. Handling fully practical models with time-dependent parameters and multiple additional components (extra normalizations, convolutions, gates, etc.) is, to the best of our knowledge, beyond current theory, and we explicitly leave this as future work.

---

> ### Author Response · Authors · 2025-11-21
> **Reply [2/3]**
>
> **Q4. The motivation for focusing on asymptotic analysis as T -> $\infty$ could be elaborated upon [...]**
>
>
> **Answer Q4.** Although our analysis is formulated in the limit $T\to\infty$, the results should be interpreted as **finite–but–large depth** statements. Modern sequence models often have large depth, and our bounds are explicit: for instance, our estimates show that the error decays like $\exp(-C_0 T)$. Thus, if one wants the error to be below a tolerance $\varepsilon$, it suffices to take the depth $T \sim \ln(\varepsilon^{-1})$. In this sense, the infinite-depth formulation is simply a mathematically clean way to capture the behavior of sufficiently deep but finite networks.
>
> Of course, many delicate questions remain about the “middle’’ regime at intermediate depths. Our work provides a first structural understanding of this regime: if the tokens are initially arranged in a random configuration, then our sign-definite coupling result implies that the token trajectories rapidly become confined to a cone (in high dimensions). Once this geometric confinement occurs, all tokens converge exponentially to the first token. We view this as a striking and qualitatively new phenomenon that is rigorously established in our analysis.
>
> Taken together, these results show that increasing the depth cannot produce new qualitative behaviors beyond a certain threshold, and that the stabilization of token dynamics is both mathematically provable and practically relevant for deep sequence models.
>
>
>
> **Q5. The purpose and practical relevance of Section 4.2 could be clarified [...]**
>
> **Answer Q5.** We appreciate the request for clarification. The purpose of Section 4.2 is not to propose a new state-of-the-art training technique, but to provide an empirical validation of our theoretical prediction: in Mamba-like models, the first-token attractor effect has measurable consequences on representation learning, even in settings where token order carries no semantic meaning (e.g., image patches).
>
> The improvements in Table 1 are indeed modest, but they are:
> (i) consistent across all runs,
> (ii) observed over 300 epochs, and
> (iii) aligned with the theoretical mechanism in Section 3 predicting that diversifying the identity of the first token should improve generalization by exploring a larger subset of the representation space.
> Thus, the role of Section 4.2 is to demonstrate that the theoretical effect we prove is real and detectable in practice, even with a simple intervention.
>
> **Q6. The explanation of Figure 1 could be made clearer [...]**
>
> **Answer Q6.** By *most trajectories converge* we specifically mean that *more than 90\%* of the simulated trajectories satisfy $\|x_1(t) - (\pm e_d)\|_2 < 10^{-3}$ for some coordinate unit vector $e_d$ after integrating the dynamical system up to time $t = 1000$. We will add this quantitative convergence criterion to the caption of Figure 1. Detailed experimental settings are provided in the caption of the figure. We have made the set up more explicitly in the revised version of the paper.
>
>
> **Q7. The paper could benefit from extending Figure 2 to additional channels and layers [...]**
>
> **Answer Q7.** We believe that adding more layers and channels to Figure 2 is not necessary: we have inspected many other channels and depths, and they exhibit the same qualitative pattern as the ones shown. The four panels we present already make the emergence of the first-token sink effect visually clear and easy to interpret, without overloading the figure.
>
> Regarding experimental details, the full setting is already provided in Supplementary Section C.1, where we describe the dataset, model configuration, inference protocol, and implementation details.

---

> ### Author Response · Authors · 2025-11-21
> **Reply [3/3]**
>
> **Q8. Equations 1619-1620 may contain a typographical error. The authors are kindly requested to verify and, if necessary, correct this in the final version.**
>
>
> **Answer Q8.** We have carefully rechecked the derivation and confirmed that there is no typographical error in the mathematical expressions themselves. The dash and spacing appearing in the draft were introduced solely for clearer line breaking and presentation in the LaTeX source. We acknowledge that this may create the impression of a typo, and we will remove or adjust this formatting in the final version to avoid any potential confusion.
>
> For completeness, we briefly outline the steps. To obtain the equation on line 1617, we differentiate
> $$f = x_2(t)\cdot x_1,$$
> which gives
> $$f' = x_2'(t)\cdot x_1.$$
> We then substitute the evolution equation for $x_2$ on line 1611 and dot both sides with $x_1$. The vector on the right-hand side,
> $$x_1 - (x_1\cdot x_2)x_2,$$
> when dotted with $x_1$, yields
> $$x_1\cdot x_1 - (x_1\cdot x_2)(x_2\cdot x_1) = 1 - |x_1\cdot x_2|^2.$$
> After replacing $x_2\cdot S_{BC}x_1$ by $g$, this gives the equation on line 1617.
>
> For the quantity $g = x_2\cdot S_{BC}x_1$, differentiating in time gives
> $$g' = \partial_t x_2 \cdot S_{BC}x_1.$$
> We again use the evolution equation on line 1611 and dot both sides with $S_{BC}x_1$. The same vector $x_1 - (x_1\cdot x_2)x_2$ now satisfies
> $$x_1\cdot S_{BC}x_1 - (x_1\cdot x_2)(x_2\cdot S_{BC}x_1),$$
> and after substituting the notations $f$ and $g$, we obtain the equation on line 1619.
>
> We hope this clarification confirms that the expressions in lines 1617–1620 are correct, and we will ensure that the final version removes any formatting that could be mistaken for a typographical error.
>
>
> References
>
> [A] Rajaraman, N., Bondaschi, M., Makkuva, A. V., Ramchandran, K., & Gastpar, M. (2024). Transformers on markov data: Constant depth suffices. Advances in Neural Information Processing Systems, 37, 137521-137556.
>
> [B] Ekbote, C., Makkuva, A. V., Bondaschi, M., Rajaraman, N., Gastpar, M., Lee, J. D., & Liang, P. P. (2025). What one cannot, two can: Two-layer transformers provably represent induction heads on any-order Markov chains. In Proceedings of the 39th Annual Conference on Neural Information Processing Systems (NeurIPS 2025).
>
> ---
> We thank the reviewer for the constructive feedback and thoughtful suggestions. If our revisions and responses have resolved the concerns, we would respectfully request that the overall score be raised accordingly. We remain fully open to further discussion in the next stage of the review process.

---

### Official Review · Reviewer_2BGT · 2025-11-04

**Soundness:** 3
**Presentation:** 3
**Contribution:** 1
**Rating:** 4
**Confidence:** 4

**Summary:**

The paper at hand extends the analysis of Karagodin et al. (2024) to Mamba-based Language models. The authors effectively start from equations (6, 7, 8), defining a dynamical system, and then study the dynamics of the first token, comparing it to those of the subsequent tokens in a causal model. After characterizing the sink effect, the authors propose a reordering strategy that improves performance on Vision Mamba models slightly.

**Strengths:**

0) The paper is easy to read and had good notation.

1) The paper is easily accessible yet not boring or trivial for researchers familiar with the topic.

2) The theoretical results well align with the experimental results.

**Weaknesses:**

I think this paper could use an additional round of polishing and could benefit from looking at the bigger picture:

0) For readers that are familiar with Karagodin et al. 2024, this paper looks a bit incremental: basically, what changes here is the structure of the attention matrix -- yet the formalism and the main idea for this analysis is not novel.

1) Looking only at Mamba is a bit limiting at this stage -- one is left wondering what happens to other linear attention mechanisms, and specifically about the relation with the discussion in https://arxiv.org/pdf/2410.10781 (section 7.4) about how attention mechanisms affect the sink effect differently. To summarize, the scope here is very limited, and there is no precise comparison to previous works.

2) The reordering strategy proposed by the authors is effective to some extent, yet it is in some way not really satisfying in the context of this paper. I believe a much better way to wrap up the work is indeed (as above written) to compare different types of attention operations and the role those have on token dynamics.

3) This is a critique that would also apply to Karagodin et al.: I believe the setup is really a bit simplistic. In particular, Mamba can have a few normalization layers (pre and post), has skip connections, convolutions and gates. None of this is present in the analysis.

**Questions:**

0) What challenge or interesting technical aspect does your analysis have that would justify publishing your work?

1) What differs in the analysis, precisely, compared to the attention case?

2) Why is the result surprising? Does this also apply to architectures such as Gated DeltaNet?

---

> ### Author Response · Authors · 2025-11-21
> **Reply [1/2]**
>
> Thank you for your thoughtful review and valuable feedback. Below we address your concerns.
>
> **W0. For readers that are familiar with Karagodin et al. 2024, this paper looks a bit incremental: basically, what changes here is the structure of the attention matrix -- yet the formalism and the main idea for this analysis is not novel.**
>
> **Q1. What challenge or interesting technical aspect does your analysis have that would justify publishing your work?**
>
> **Q2. What differs in the analysis, precisely, compared to the attention case?**
>
> **Q3a. Why is the result surprising?**
>
>
> **Answer W0-Q1-Q2-Q3a.** We *strongly disagree* with the assessment that our contribution is merely incremental relative to Karagodin et al. (2024). Although our work is inspired by theirs, the Mamba dynamics introduce genuinely new mathematical challenges that do not arise in the Transformer setting.
>
> First, the effective coupling coefficients in Mamba are ***not sign-definite***. This loss of positivity is created by the additional term $\langle x_\ell, S_{BC} x_j \rangle$, which can take either sign for general $x_\ell, x_j$. In the Transformer case analyzed by Karagodin et al., all analogous coefficients are automatically positive, and we explicitly verify this when rederiving their result in Theorem B.16. The techniques in Karagodin et al. fundamentally rely on this positivity and therefore do not carry over to our setting.
>
> The core technical novelty of our work is the construction of a suitable ambient domain and the proof that all Mamba coefficients remain inside this “box’’ for all time. This requires new geometric ideas together with a nontrivial exit-time argument (see page 31). The full dynamical system governing these coefficients is derived in Lemma B.10 and ***has no direct analogue in the Transformer analysis***.
>
> ***The absence of a sign structure also makes the nonlinear convergence analysis substantially more difficult than in the 2024 paper***. To overcome this, we introduce a new multi-step strategy:
>
> - We first analyze an auxiliary system with a positively coupled structure (with deliberately simplified data).
>
> - We then prove that the true Mamba coefficients eventually become sign-definite, which lets us rigorously connect the auxiliary system back to the real dynamics.
>
> - Finally, we establish exponential convergence using PDE-inspired techniques, in particular sharp Grönwall-type estimates. This is mathematically distinct from the 2024 approach, which relies on dynamical-systems and center-manifold tools. Our method is more explicit and quantitative and yields concrete bounds that are not available in the prior Transformer analysis.
>
> Moreover, ***our linearized analysis introduces an additional conceptual innovation: rather than studying the full Jacobian, we isolate the most unstable direction and track the associated eigenvalue directly***. This directional approach avoids a full Jacobian analysis and leads to a cleaner and more transparent proof. We view this idea as independently useful for future work on token dynamics and related nonlinear systems.
>
> In summary, the structural differences in the dynamics, the loss and later recovery of sign-definiteness, the new geometric “box’’ construction with exit-time control, the PDE-based convergence argument, and the directional linearization around the equilibrium together constitute a substantially different and technically stronger analysis than in Karagodin et al. (2024). For these reasons, we do not consider our contribution to be incremental.
>
> **W1. Looking only at Mamba is a bit limiting at this stage -- one is left wondering what happens to other linear attention mechanisms, and specifically about the relation with the discussion in https://arxiv.org/pdf/2410.10781 (section 7.4) about how attention mechanisms affect the sink effect differently [...]**
>
> **Answer W1.** Our analysis is not limited to Mamba. In Section 3.3, we develop a general framework for autoregressive sequence models and show that the attracting effect of the first token extends to a broad class of architectures, including state space models, causal Transformers, and classical time-series models.
>
> Regarding [A], the scope is fundamentally different: [A] provides observations, while we provide theoretical proofs. In particular, [A] provides empirical observations of the sink phenomenon under different attention mechanisms, whereas our work gives the first rigorous theoretical proof of the sink effect they observe. We will make this distinction and the generality of our results clearer in the revised version.
>
> [A] Xiangming Gu et al., When Attention Sink Emerges in Language Models: An Empirical View. ICLR 2025. https://arxiv.org/pdf/2410.10781

---

> ### Author Response · Authors · 2025-11-21
> **Reply [2/2]**
>
> **W2. The reordering strategy proposed by the authors is effective to some extent, yet it is in some way not really satisfying in the context of this paper. I believe a much better way to wrap up the work is indeed (as above written) to compare different types of attention operations and the role those have on token dynamics.**
>
> **Answer W2.** Our goal with the reordering strategy is to give a practical, theory-driven mitigation that follows directly from our dynamical analysis, not to provide a broad empirical comparison of attention mechanisms. A systematic comparison of different attention types is interesting, but it is outside the scope of this paper, whose main focus is to provide the first rigorous analysis of the sink effect.
>
> Concretely, for Mamba we propose randomly reordering tokens during training to diversify clustering around a small set of limit points and thereby improve performance. This is intended as a practical way to alleviate the sink phenomenon derived from our theory, not as a full study of alternative attention operations.
>
>
> **W3. This is a critique that would also apply to Karagodin et al.: I believe the setup is really a bit simplistic. In particular, Mamba can have a few normalization layers (pre and post), has skip connections, convolutions and gates. None of this is present in the analysis.**
>
>
> **Answer W3.** While your concern about the simplified setting used in our theory is valid, it is standard in theoretical work to adopt simplified architectures in order to make rigorous analysis feasible, while still retaining enough structure to capture the key behaviors observed in practice. Several high-quality related works by leading researchers follow this approach, for example:
>
> [B] Karagodin et al., Clustering in Causal Attention Masking, NeurIPS 2024.
>
> [C] Geshkovski et al., The Emergence of Clusters in Self-Attention Dynamics, NeurIPS 2024.
>
> [D] Geshkovski et al., A Mathematical Perspective on Transformers, Bulletin of the AMS 2024.
>
> [E] Vo et al., Demystifying the Token Dynamics of Deep Selective State Space Models, ICLR 2025 (spotlight).
>
> All above papers using time-independent parameters and one key components (either attention in Transformer or S6 in Mamba).
>
> Our results generalize the findings in [E] by incorporating layer normalization and studying token dynamics on high-dimensional spheres. We further extend these results to general autoregressive sequence models, which we view as a substantial and nontrivial generalization.
>
> Handling models with time-dependent parameters and additional practical components (such as multiple normalization layers, convolutions, or gating mechanisms) remains extremely challenging, and to the best of our knowledge, no existing theoretical work has achieved this. We agree that this is an important direction and leave it for future research.
>
> **Q3b. Does this also apply to architectures such as Gated DeltaNet?**
>
> **Answer Q3.** Our theoretical findings do apply to Gated DeltaNet [F], provided that the additional control-state decay $\alpha_t$ is time-independent. The core difference between the Mamba-2 setting we study and Gated DeltaNet is precisely this extra control-state decay term; under a time-independent $\alpha_t$, our analysis and results continue to hold.
>
> If $\alpha_t$ is time-dependent, the system becomes non-autonomous, and in that regime we do not currently know whether our theorems remain valid. A rigorous treatment of this non-autonomous case, together with more general time-dependent parameters, is an interesting but technically challenging direction that we leave for future work.
>
> [F] Songlin Yang et al. Gated Delta Networks: Improving Mamba2 with delta rule. ICLR 2025.
>
> ---
>
> We thank the reviewer for the constructive feedback and thoughtful suggestions. If our revisions and responses have resolved the concerns, we would respectfully request that the overall score be raised accordingly. We remain fully open to further discussion in the next stage of the review process.

---

### Meta-Review · Area_Chair_68bt · 2026-01-07

**Summary:**

This work analyzes state updates in solvable Mamba models from a dynamical systems perspective. The theory sounds good, and concrete results are provided. However, all reviewers give reject-side evaluations. In particular,

*Similar idealized solvable models have been explored in the past few years, especially for self-attention, and their importance has been recognized in the research community. Just applying them to Mamba provides a limited contribution to both theory and practice. More fine-grained investigation into the justification of the simplifying assumptions or settings such as infinite T would be required at the current stage.

*To increase the contribution, more novel and quantitative results for general autoregressive sequence models would be desirable.
To reach the acceptance line of the current ICLR, the current manuscript would require further fundamental revisions, and is unavoidably evaluated as rejection.

**Reviewer Concerns:**

The following concerns are outstanding:

**Reviewer 2BGT**
- 1. Looking only at Mamba is a bit limiting at this stage...
- 2. ... much better way to wrap up the work is indeed (as above written) to compare different types of attention operations and the role those have on token dynamics.

**Reviewer FDLD**
- The paper makes several assumptions that seem insufficiently supported

**Reviewer ZK7P**
- 2. The range of applications is limited...

**Reviewer Scores:**

The reviewers are fairly confident on their claims and I expect no score changes.

---

### Decision · Program_Chairs · 2026-01-26

Reject